# Cryo-EM structures of mitochondrial respiratory complex I from *Drosophila melanogaster*

**Ahmed-Noor A Agip[†], Injae Chung[†], Alvaro Sanchez-Martinez[†], Alexander J Whitworth\*, Judy Hirst\***

The Medical Research Council Mitochondrial Biology Unit, University of Cambridge, The Keith Peters Building, Cambridge Biomedical Campus, Cambridge, United Kingdom

**\*For correspondence:**
a.whitworth@mrc-mbu.cam.ac.uk (AJW);
jh@mrc-mbu.cam.ac.uk (JH)

[†]These authors contributed equally to this work

**Competing interest:** The authors declare that no competing interests exist.

**Abstract** Respiratory complex I powers ATP synthesis by oxidative phosphorylation, exploiting the energy from NADH oxidation by ubiquinone to drive protons across an energy-transducing membrane. *Drosophila melanogaster* is a candidate model organism for complex I due to its high evolutionary conservation with the mammalian enzyme, well-developed genetic toolkit, and complex physiology for studies in specific cell types and tissues. Here, we isolate complex I from *Drosophila* and determine its structure, revealing a 43-subunit assembly with high structural homology to its 45-subunit mammalian counterpart, including a hitherto unknown homologue to subunit NDUFA3. The major conformational state of the *Drosophila* enzyme is the mammalian-type 'ready-to-go' active resting state, with a fully ordered and enclosed ubiquinone-binding site, but a subtly altered global conformation related to changes in subunit ND6. The mammalian-type 'deactive' pronounced resting state is not observed: in two minor states, the ubiquinone-binding site is unchanged, but a deactive-type π-bulge is present in ND6-TMH3. Our detailed structural knowledge of *Drosophila* complex I provides a foundation for new approaches to disentangle mechanisms of complex I catalysis and regulation in bioenergetics and physiology.

## Editor's evaluation

This important article advances our understanding of respiratory complex I. The cryoEM data are convincing and the interpretation of different conformational states will stimulate discussions in the field. The work introduces *Drosophila melanogaster* as a model organism to study respiratory complex I and will be of interest to researchers studying respiratory enzymes, the evolution of respiration and mitochondrial diseases.

## Introduction

Mitochondrial complex I (NADH:ubiquinone oxidoreductase) is a crucial enzyme in cellular metabolism, central to NAD$^+$ homeostasis, respiration, and oxidative phosphorylation, and a key contributor to the production of cellular reactive oxygen species (ROS) (**Hirst, 2013**; **Parey et al., 2020**). By catalyzing NADH oxidation in the mitochondrial matrix coupled to ubiquinone reduction in the inner membrane, it regenerates the oxidised NAD$^+$ pool to sustain crucial metabolic processes, including the tricarboxylic acid cycle and β-oxidation, and provides reducing equivalents to the downstream complexes of the electron transport chain. The energy from NADH:ubiquinone oxidoreduction is harnessed to transport four protons across the inner membrane (**Jones et al., 2017**), supporting the proton motive force (Δp) that drives ATP synthesis and transport processes. These central roles of complex I in both

metabolism and oxidative stress make complex I dysfunctions, induced by genetic, pharmacological, and environmental factors, some of the most frequent primary causes of mitochondrial diseases, as well as a contributor to many socially and economically important diseases common in ageing populations (*Fassone and Rahman, 2012*; *Fiedorczuk and Sazanov, 2018*; *Padavannil et al., 2022*). For example, ROS production by complex I operating in reverse, during 'reverse electron transfer' (RET, Δp-driven ubiquinol:NAD⁺ oxidoreduction) (*Pryde and Hirst, 2011*), is a major contributor to the tissue damage that occurs in strokes and heart attacks, during ischaemia-reperfusion (IR) injury (*Chouchani et al., 2016*; *Chouchani et al., 2014*; *Dröse et al., 2016*; *Yin et al., 2021*).

Mammalian complex I is a 1 MDa asymmetric assembly of 45 subunits, encoded on both the nuclear and mitochondrial genomes (*Hirst, 2013*; *Hirst et al., 2003*; *Zhu et al., 2016*). Fourteen of them (seven nuclear and seven mitochondrial) are the core subunits conserved in all complex I homologues that are essential for catalysis, whereas the other 31 subunits are supernumerary subunits that are involved in enzyme assembly, stability, and regulation, or that have independent roles within the cell (*Hirst et al., 2003*; *Padavannil et al., 2022*; *Zhu et al., 2016*). Bioinformatic analyses have indicated how the cohort of supernumerary subunits has been augmented gradually throughout the evolution of the eukaryotic complex (*Gabaldón et al., 2005*), and an increasing range of structural analyses of different species of complex I now illustrates the diversity of the supernumerary subunit cohorts that have developed in different eukaryotic lineages (*Klusch et al., 2021*; *Parey et al., 2021*; *Soufari et al., 2020*; *Zhou et al., 2022*).

For mammalian complex I, the form of the enzyme most relevant in medicine, single-particle electron cryomicroscopy (cryo-EM) has yielded detailed structural information on multiple different states of the complex (*Chung et al., 2022a*; *Kampjut and Sazanov, 2022*; *Parey et al., 2020*). However, detailed structure–function studies are limited for the mammalian enzyme due to substantial challenges in creating and studying genetic variants in representative mammalian model systems, such as mouse. Whereas simpler model systems, such as α-proteobacteria or yeast species (*Jarman et al., 2021*; *Jarman and Hirst, 2022*; *Kravchuk et al., 2022*; *Parey et al., 2019*), allow far greater opportunities for genetic studies, the protein compositions of their complex I vary substantially from the mammalian enzyme, fail to recapitulate key characteristics and behaviour of the mammalian complex such as the 'active/deactive transition' (*Babot et al., 2014*; *Kotlyar and Vinogradov, 1990*; *Maklashina et al., 2003*; *Vinogradov, 1998*), and the physiological environments in which the variant complexes can be studied are very restricted. Most relevant here, the active and deactive states of mammalian complex I are two biochemically and structurally characterised resting states of the complex (*Agip et al., 2018*; *Blaza et al., 2018*; *Chung et al., 2022a*; *Chung et al., 2022b*; *Zhu et al., 2016*): the 'active' ready-to-go resting state and the 'deactive' pronounced resting state. They differ both in their global conformations and in the status of local structural features. In particular, the ubiquinone-binding site in the active state is fully enclosed and sealed, whereas in the deactive state disorder in the enclosing loops opens the site to the matrix (*Agip et al., 2018*; *Blaza et al., 2018*; *Chung et al., 2022a*; *Chung et al., 2022b*; *Zhu et al., 2016*). The active and deactive resting states have also been referred to as the 'closed' and 'open' states of the mammalian enzyme on the basis of changes in the apparent angle between their membrane and hydrophilic domains (*Kampjut and Sazanov, 2020*). Finally, we note that there is currently substantial controversy about the biochemical and physiological relevance of the open states of the mammalian complex (*Chung et al., 2022a*), which have recently been proposed to include not only the deactive resting state but also on-cycle catalytic intermediates (*Kampjut and Sazanov, 2020*; *Kravchuk et al., 2022*).

The fruit fly, *Drosophila melanogaster*, is a powerful genetically tractable model organism for metazoa. *Drosophila* encodes a complex I with a composition that closely resembles that of the mammalian complex (*Gabaldón et al., 2005*; *Rhooms et al., 2020*), with clear homologues to 42 of the 44 mammalian subunits identified. Therefore, in addition to providing an additional model system for studying the mechanism of complex I catalysis (also accessible in simpler unicellular models), variants in *Drosophila* complex I can be studied for their effects on regulation and assembly (*Cho et al., 2012*; *Garcia et al., 2017*; *Murari et al., 2020*). Furthermore, *Drosophila* can potentially be exploited to investigate features of complex I function that are observed for mammalian complex I, but not universal features of the enzyme in simpler organisms, such as the active/deactive transition, RET, and the involvement of complex I in supercomplexes (*Garcia et al., 2017*; *Scialò et al., 2016*; *Shimada et al., 2018*). For instance, studies in *Drosophila* have proposed that RET-ROS increase

lifespan (*Scialò et al., 2016*) and *Drosophila* are remarkably resistant to hypoxic or anoxic exposure (*Haddad, 2006*; *Zhou and Haddad, 2013*), which might provide insights into pathological mechanisms of RET-mediated IR injury. Furthermore, with substantial tissues, such as indirect flight muscles, highly enriched with mitochondria, *Drosophila* represent an attractive animal model for the analysis of basic mitochondrial biology, offering a complex physiological system for the generation and study of complex I genetic variants at the whole organism or tissue-specific level, as well as the involvement of complex I in differing physiological conditions.

To date, no detailed molecular studies of *Drosophila* complex I have been pursued to confirm its structural and functional similarity with the mammalian enzyme, or exploit its potential as a metazoan model system. Therefore, we sought here to structurally and biochemically evaluate *Drosophila* as a model system for mammalian complex I. We determine structures for three distinct conformational states of the *Drosophila* enzyme and compare them to well-characterised resting states of the mammalian complex, leading to new insights into the mammalian active/deactive transition and enhancing understanding of the conformational link between the ubiquinone-binding site and the proximal membrane domain. We thus present detailed knowledge of *Drosophila* complex I at the molecular level and confirm and define its relationships to the mammalian enzyme.

## Results

### The 43-subunit structure of *Drosophila* complex I

Complex I was isolated from mitochondrial membranes prepared from whole adult *Drosophila* by detergent extraction from the membrane followed by anion-exchange and size-exclusion chromatography, according to a small-scale protocol developed previously for mammalian complex I (*Agip et al., 2018*). The complex eluted from the size-exclusion column in a homogeneous peak consistent with the expected ~1 MDa mass of the monomeric complex (*Figure 1—figure supplement 1*). The highest concentration peak fraction (3.4 mg mL$^{-1}$), which exhibited an NADH oxidoreductase activity comparable to mammalian complex I of 7.3 ± 0.3 μmol min$^{-1}$ mg$^{-1}$ (ca. 120 NADH s$^{-1}$), was collected and frozen onto thiol-modified gold cryo-EM grids (*Blaza et al., 2018*; *Meyerson et al., 2015*; *Russo and Passmore, 2014*). The grids were imaged using a 300 KeV Titan Krios microscope equipped with a Gatan K2 camera and GIF Quantum energy filter (*Table 1*), and 63,471 particles images were selected and processed using *RELION* (*Zivanov et al., 2020*; *Zivanov et al., 2018*) into three major classes (*Figure 1—figure supplement 2*). The highest resolution map reached an estimated global resolution of 3.3 Å with consistent local resolution, and the two smaller subclasses reached estimated global resolutions of 3.7 and 4.0 Å (*Figure 1—figure supplement 3*). Example densities are shown in *Figure 1—figure supplements 4–6*.

*Figure 1* shows the overall structure of *Drosophila* complex I, which consists of 43 subunits: 14 core subunits (*Figure 1a*) and 29 supernumerary subunits (*Figure 1b*). The 14 core subunits comprise the canonical heart of the enzyme that is conserved throughout all species of complex I, with the core subunits of the *Drosophila* and mammalian (bovine, PDB ID: 7QSK; *Chung et al., 2022b*) enzymes exhibiting an overall root mean square deviation (RMSD) of 1.065 Å. The 29 supernumerary subunits all correspond to supernumerary subunits found in mammalian complex I, confirming the close relationship between them. However, two supernumerary subunits present in mammalian complex I are absent from the *Drosophila* complex (*Figure 1c*): subunits NDUFC1 and NDUFA2 (to aid comparisons to the mammalian enzyme, we use the human nomenclature throughout; see, e.g., *Rhooms et al., 2020* for a list of the corresponding gene names in *Drosophila*). NDUFC1 is a short, single transmembrane helix (TMH)-containing subunit in the membrane domain that is peripherally associated with the mammalian complex through its interaction with subunit NDUFC2, and subunit NDUFA2 binds to subunit NDUFS1 at the top of the hydrophilic domain in the mammalian complex. The absence of NDUFC1 was expected since no orthologue was identified in the *Drosophila* genome by bioinformatic analyses (*Gabaldón et al., 2005*), and in *Drosophila* the N-terminus of NDUFC2 is displaced by the C-terminal extension of NDUFA11 (see *Figure 1—figure supplement 5*). Based on the same bioinformatic analyses, subunit NDUFA3 was also expected to be absent, but a matching subunit (Dmel gene *CG9034*) was detected by mass spectrometry in our preparation (see 'Materials and methods') and is clearly present in our density map in the location of mammalian-NDUFA3 in the membrane domain (see *Figure 1—figure supplement 6*). However, the sequence homology is weak and the structures

**Table 1.** Cryo-EM data collection, refinement, and validation statistics for the three states of *Drosophila* complex I.

| | *Drosophila melanogaster* complex I dataset | | |
|---|---|---|---|
| **Data collection and processing** | | | |
| Magnification | 130,000 | | |
| Voltage (kV) | 300 | | |
| Electron exposure (e$^-$/Å$^2$) | 42 | | |
| Defocus range (μm) | –1.0 to –2.0 | | |
| Nominal pixel size (Å) | 1.07 | | |
| Calibrated pixel size (Å) | 1.048 | | |
| Symmetry imposed | C1 | | |
| Initial particle images (no.) | 194,538 | | |
| Final particle images (no.) | 63,471 | | |
| | | | |
| **Classes** | *Dm*1 [Active] | *Dm*2 [Twisted] | *Dm*3 [Cracked] |
| | EMD-15936 | EMD-15937 | EMD-15938 |
| | PDB-8B9Z | PDB-8BA0 | |
| Final particle images (no.) | 37,608 | 12,343 | 13,520 |
| Map resolution (Å) | | | |
| FSC threshold: 0.143 | 3.28 | 3.68 | 3.96 |
| Map resolution range (Å) | 2.98–6.19 | 3.33–9.40 | 3.51–11.35 |
| Map sharpening *B*-factor (Å$^2$) | 0 | 15 | 20 |
| | | | |
| **Model statistics** | | | |
| Initial model (PDB ID) | 6G2J | 6G2J | |
| Model resolution (Å) | | | |
| FSC threshold: 0.5 | 3.41 | 3.92 | |
| Model composition | | | |
| Non-hydrogen atoms | 66,970 | 65,912 | |
| Protein residues | 8,178 | 8,136 | |
| Ligands | 39 | 22 | |
| Average *B*-factors (Å$^2$) | | | |
| Protein | 99 | 110 | |
| Ligand | 103 | 115 | |
| Root mean square deviations | | | |
| Bond lengths (Å) | 0.006 | 0.007 | |
| Bond angles (°) | 0.726 | 0.783 | |
| MolProbity score | 1.60 | 2.04 | |
| All-atom clash score | 6.24 | 10.90 | |
| EMRinger score | 3.18 | 1.68 | |
| Rotamer outliers (%) | 0.00 | 0.00 | |

*Table 1 continued on next page*

*Table 1 continued*

| Classes | *Dm*1 [Active] | *Dm*2 [Twisted] | *Dm*3 [Cracked] |
|---|---|---|---|
| Ramachandran plot | | | |
| Favoured (%) | 96.18 | 91.97 | |
| Allowed (%) | 3.80 | 8.01 | |
| Outliers (%) | 0.02 | 0.02 | |

of the two proteins diverge in the C-terminal membrane-extrinsic domain, with the obtuse-angled 'turn' that follows the TMH in the mammalian protein sterically blocked by the marginally extended C-terminal TMH of ND1 in *Drosophila*. NDUFA2, which has a characteristic thioredoxin fold and is widely conserved in eukaryotic complex I, is surprisingly absent from our *Drosophila* structure despite a highly conserved homologue in the *Drosophila* genome (*Gabaldón et al., 2005*). However, NDUFA2 interacts with only subunit NDUFS1 in the mammalian complex, and inspection of the (otherwise highly conserved) region of interaction in the *Drosophila* enzyme shows local disorder in a specific helix in the *Drosophila* NDUFS1 subunit (residues 673–684) that binds NDUFA2 in the mammalian enzyme. Although this result suggests NDUFA2 is associated with *Drosophila* complex I in vivo but has been lost during enzyme purification, detailed transcriptomic analyses (*Brown et al., 2014*; *Leader et al., 2018*) show that NDUFA2 (*Drosophila* ND-B8) expression is restricted principally to the male germline, and therefore the NDUFA2 protein is unlikely to be a constitutive component of complex I in somatic tissues.

Overall, *Drosophila* complex I is remarkably similar in its composition and structure to the mammalian enzyme, underlining expectations of the value of *Drosophila* as a model system for complex I research. Only further minor differences are present in some of the subunits (*Figure 1—figure supplement 5*). ND5-TMH1 is absent from the *Drosophila* subunit: although ND2, ND4, and ND5 have a canonical 14-TMH core structure, truncation of the N-terminal TMHs appears tolerated, consistent with them lacking specific catalytically active residues or features, and demonstrated by the 11-TMH form of subunit ND2 in bilateria that lacks the three N-terminal TMHs found in lower organisms (*Birrell and Hirst, 2010*). In addition, the structures of supernumerary subunits NDUFB6 and NDUFB1 are noticeably different in the *Drosophila* enzyme (*Figure 1—figure supplement 5*). Notably, all the substantial differences in the membrane domain (absence of NDUFC1 and ND5-TMH1, variations in NDUFA3, NDUFB6 and NDUFB1, extension of NDUFA11) are located on the 'right' side of the boot-shaped enzyme, perhaps because there is less evolutionary pressure on the right side than on the left, where interactions with complexes III and IV are central to the stabilisation of respiratory chain supercomplexes (*Milenkovic et al., 2017*).

## Three distinct states of *Drosophila* complex I

Cryo-EM particle classification identified three distinct states in our preparation of *Drosophila* complex I, which we refer to as *Dm*1, *Dm*2, and *Dm*3 (*Figure 1—figure supplement 2*). The *Dm*1 class is the dominant class, containing ~60% of the particles, whereas the two minor classes, *Dm*2 and *Dm*3, each contain ~20%. On a global scale (*Figure 2*), the *Dm*2 state is 'twisted' relative to the *Dm*1 state: with the two models aligned on subunit ND1 in the 'heel' of the complex, the hydrophilic and membrane domains twist in opposite directions (there is no apparent opening or closing motion between the domains). A similar twisting relationship was identified between the active and deactive resting states of mammalian complex I (*Zhu et al., 2016*). However, standard biochemical assays used to detect the presence of the mammalian deactive state did not detect any deactive *Drosophila* enzyme, even after incubation at 37°C to promote deactivation (*Figure 2—figure supplement 1*), indicating that *Dm*2 is not directly comparable to the mammalian-type deactive state. In the mammalian deactive state, the equivalent residue to ND3-Cys41 (we use *Drosophila* numbering throughout) is exposed to solution and can be derivatised by *N*-ethylmaleimide (NEM), preventing reactivation of the deactive enzyme and its return to catalysis, whereas in the active state ND3-Cys41 is buried (*Galkin et al., 2008*). Our assays suggest that either ND3-Cys41 is buried and inaccessible to derivatisation in all three *Dm*1, *Dm*2, and *Dm*3 states, or that ND3-Cys41 is exposed in one or more state that is completely inactive, being unable to either reactivate or catalyse. For the *Dm*3 state, the most obvious global feature

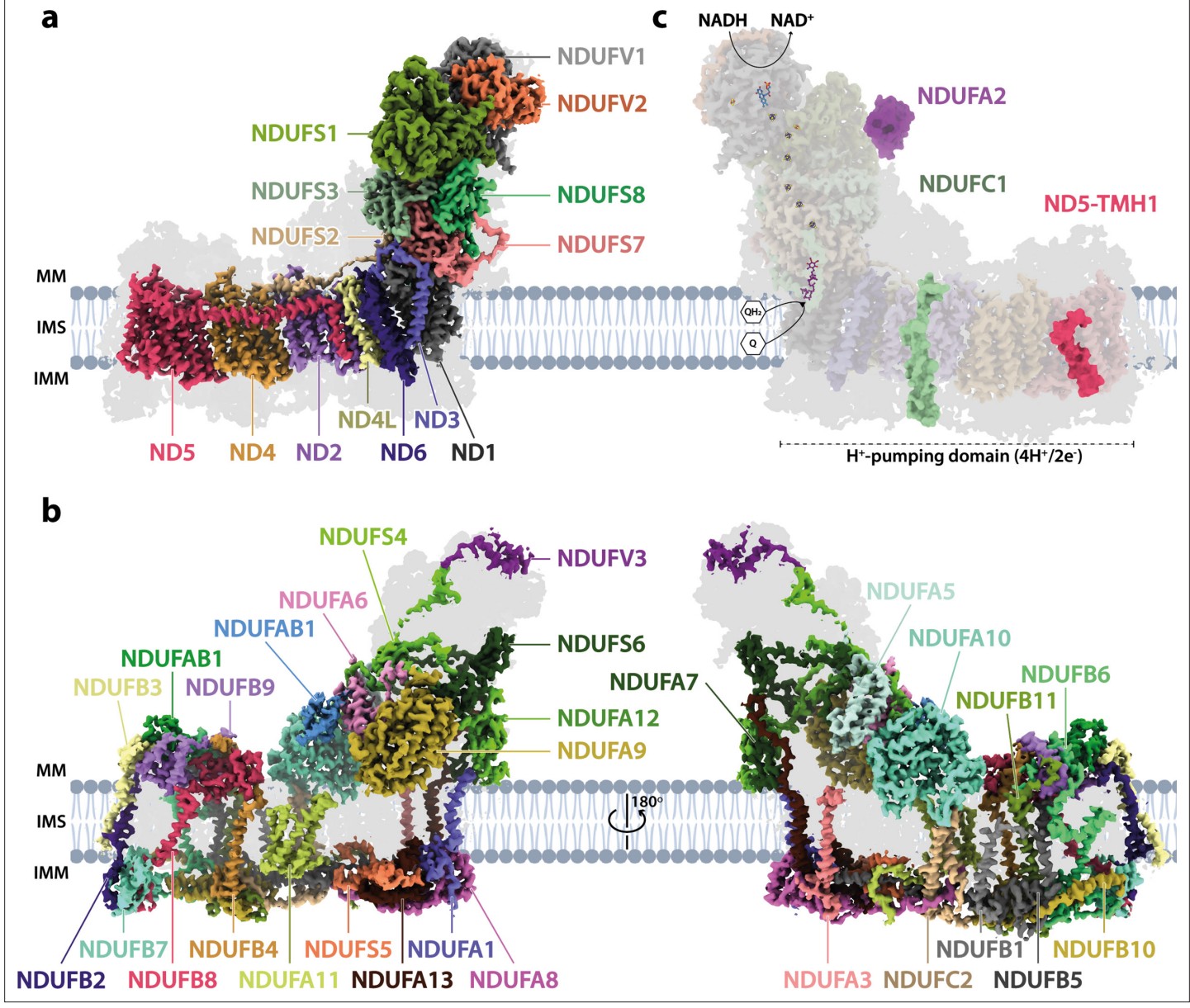

**Figure 1.** The architecture of complex I from *Drosophila melanogaster*. (**a**) The 14 core subunits are shown in colour and labelled accordingly, and the 29 supernumerary subunits are shaded in grey. (**b**) The 29 supernumerary subunits are shown in colour and labelled accordingly, and the 14 core subunits are shaded in grey. (**c**) *Drosophila* complex I shown in transparent colour (as in **a**) with NDUFA2 (purple), NDUFC1 (green), and ND5-TMH1 (red), which are absent in *Drosophila* but present in mammalian complex I, indicated in solid colour from the structure of bovine complex I (PDB ID: 7QSK) (*Chung et al., 2022b*). The NADH-binding site at the flavin mononucleotide (FMN) cofactor, iron-sulphur clusters, the ubiquinone-binding site ($Q_9$; purple), and the proton-pumping domain are indicated. All structures are of the *Dm*1 active-state *Drosophila* cryo-EM map, shown at a map threshold of 0.013 in *UCSF ChimeraX* (*Pettersen et al., 2021*). MM, mitochondrial matrix; IMS, intermembrane space; IMM, inner mitochondrial membrane; TMH, transmembrane helix; Q, ubiquinone; $QH_2$, ubiquinol.

The online version of this article includes the following figure supplement(s) for figure 1:

**Figure supplement 1.** Example of a preparation of *Drosophila* complex I.

**Figure supplement 2.** Cryo-EM data processing and particle classification.

**Figure supplement 3.** Local resolution maps, Mollweide projections, *3DFSC* plots, and Fourier shell correlation (FSC) curves for three states of *Drosophila* complex I.

**Figure supplement 4.** Cryo-EM densities and models for ligands and phospholipids observed in *Drosophila* complex I.

**Figure supplement 5.** Cryo-EM densities and models for *Drosophila*-specific subunit extensions and conformations.

**Figure supplement 6.** Cryo-EM density and model for subunit NDUFA3 in *Drosophila* complex I.

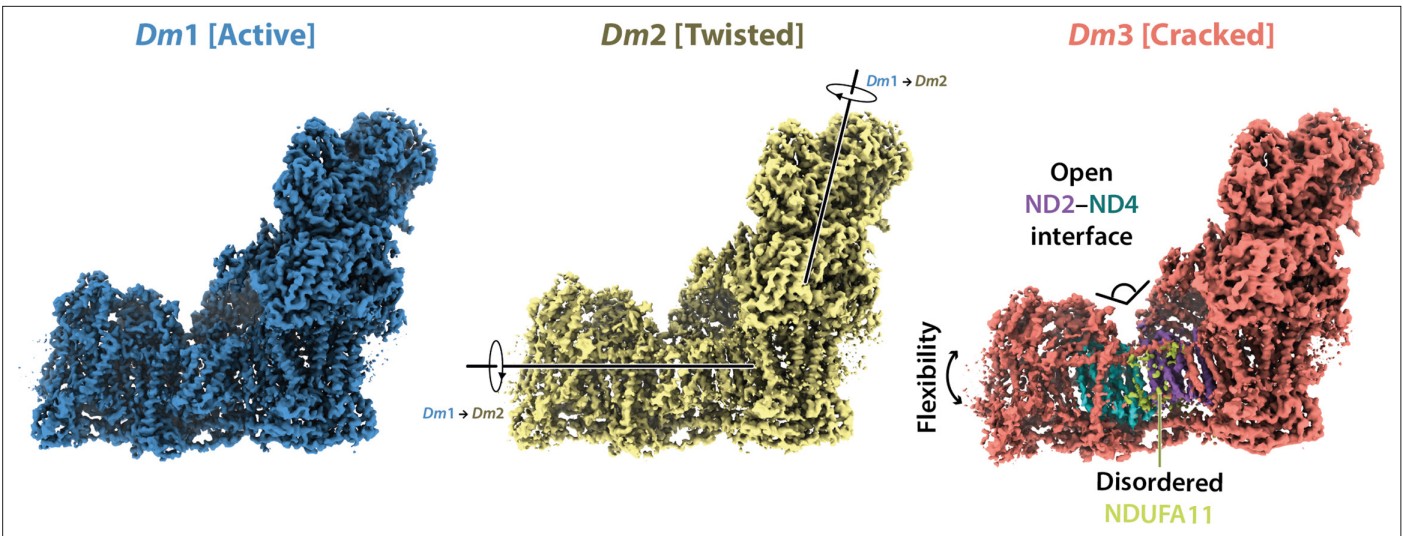

**Figure 2.** Global comparison between the three states of *Drosophila* complex I. Side views of the three *Drosophila* complex I cryo-EM maps identified by 3D classification are shown with global motions between the three states indicated. States *Dm*1 and *Dm*2 are related by a twisting motion of the hydrophilic and membrane domains about the ND1-containing 'heel' subdomain. States *Dm*2 and *Dm*3 are related by 'cracking' open of the ND2–ND4 interface in *Dm*3. Cryo-EM densities are shown at map thresholds of 0.013 (*Dm*1), 0.014 (*Dm*2), and 0.015 (*Dm*3) in *UCSF ChimeraX* (*Pettersen et al., 2021*).

The online version of this article includes the following figure supplement(s) for figure 2:

**Figure supplement 1.** The *N*-ethylmaleimide (NEM) assay does not reveal a mammalian-type deactive state for *Drosophila* complex I.

**Figure supplement 2.** Structural features of the *Dm*3 state of *Drosophila* complex I.

(*Figure 2*) is that the membrane domain appears 'cracked' at the interface between ND2 and ND4; the density for the adjacent subunit NDUFA11 is disordered, along with the adjacent N-terminus of NDUFS2 (*Figure 2—figure supplement 2*). These characteristics resemble those of the 'slack' state of bovine complex I (*Chung et al., 2022b*; *Zhu et al., 2016*), which is of uncertain biochemical and physiological relevance and which may result from destabilisation of the membrane-intrinsic domain following extraction from the membrane and delipidation by detergents during purification. To evaluate the three states of 'resting' *Drosophila* complex I further, we first focus on the *Dm*1 state and its relationship with known resting states of the mammalian enzyme.

### *Dm*1 is the active resting state of *Drosophila* complex I

In addition to differing in their global conformations, the mammalian active and deactive states are differentiated by the status of a set of local features in the core subunits (*Agip et al., 2018*; *Blaza et al., 2018*; *Chung et al., 2022a*; *Chung et al., 2022b*; *Zhu et al., 2016*). In the *Dm*1 state of *Drosophila* complex I, these features are all unambiguously in the active state (*Figure 3a*). First, ND6-TMH3 is clearly α-helical, it does not contain the π-bulge that is characteristic of the deactive state, and ND1-TMH4 is clearly in the 'bent' conformation of the active state (with Tyr149 pointing toward the E-channel), not the straight conformation of the deactive state (with Tyr149 pointing away from the E-channel). Second, the densities for the NDUFS2-β1–β2 loop that carries the His97 ligand to bound ubiquinone, the ND3-TMH1−2 loop that carries Cys41 (the biochemical marker of the mammalian active/deactive states), and the ND1-TMH5−6 loop, are all well-defined in the *Dm*1 density map and their conformations match the mammalian active-state conformations (they are not disordered as in the deactive state). ND3-Cys41, NDUFS2-His93, and ND1-Tyr134 meet in a trigonal junction at the top of ND1-TMH4 (*Grba and Hirst, 2020*), as they do in the active state (*Figure 3c*). Finally, the FRASPR motif in NDUFS7 that includes Arg119 matches its conformation in the mammalian active, not deactive, state.

Importantly, as expected from the ordered states of the loop structures that form the ubiquinone-binding site, the site is sealed and closed from the matrix (*Figure 3b and c*) as in the mammalian active state, not open to the matrix as in the deactive state (*Agip et al., 2018*; *Blaza et al., 2018*;

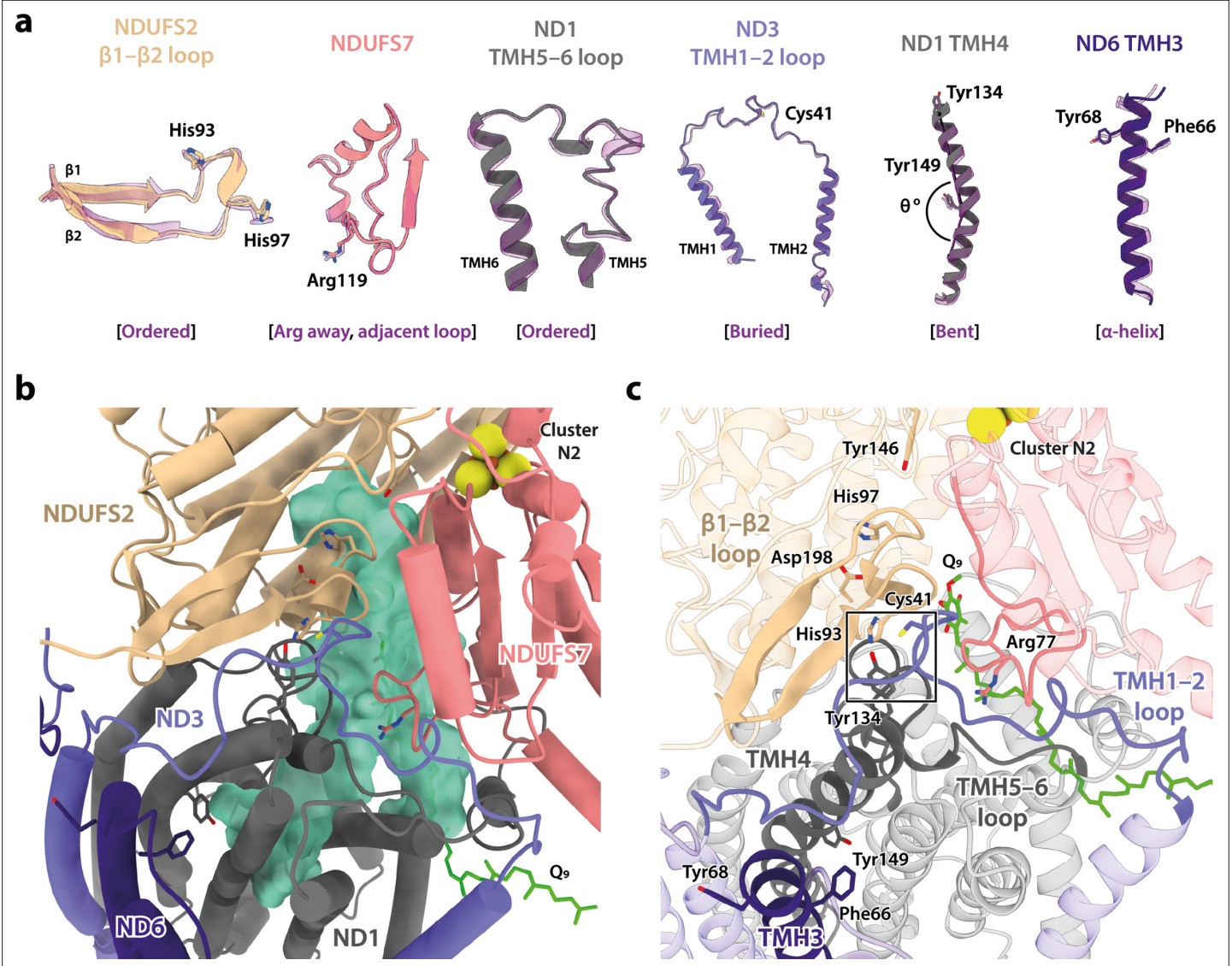

**Figure 3.** Local structural elements show that *Dm*1 is the active resting state of *Drosophila* complex I. (**a**) The local elements in the core subunits that show the *Dm*1 complex is in the active state are individually compared against an active-state bovine structure (transparent purple; PDB ID: 7QSK) (***Chung et al., 2022b***). Active state-specific key local features are indicated in square brackets. The same features are highlighted and labelled in (**b**) and (**c**), respectively, showing that subunits NDUFS2, NDUFS7, ND1, ND3, and ND6 encapsulate a fully structured and sealed Q-binding cavity (aquamarine surface; detected by *CASTp; Tian et al., 2018*) with a $Q_9$ molecule bound. The Coulomb potential density for $Q_9$ is shown in ***Figure 1—figure supplement 4***. The box in (**c**) denotes the trigonal junction (ND3-Cys41, NDUFS2-His93, and ND1-Tyr134).

The online version of this article includes the following figure supplement(s) for figure 3:

**Figure supplement 1.** Comparison of the position of the bound ubiquinone in the *Dm*1 state with the positions of ubiquinone molecules bound in other structures.

***Chung et al., 2022a***; ***Chung et al., 2022b***; ***Zhu et al., 2016***). This observation indicates that the *Dm*1 state is a catalytically competent state, ready to bind and reduce the extended and hydrophobic ubiquinone-9 or ubiquinone-10 substrate (referred to as $Q_9$ for brevity). Indeed, density for $Q_9$ is observed within the site, although the $Q_9$ is only partially inserted, with its ubiquinone-headgroup in the central section of the channel, rather than ligated to the two proton-donor ligands NDUFS2-His97 and NDUFS2-Tyr146 as required for its reduction (***Baradaran et al., 2013***; ***Tocilescu et al., 2010***). Partially inserted ubiquinones have been observed previously in several different species and states of complex I (***Gu et al., 2022***; ***Kampjut and Sazanov, 2020***; ***Kravchuk et al., 2022***; ***Parey et al., 2019***; ***Soufari et al., 2020***), but the headgroup typically sits slightly

lower down the channel than observed here, in an array of positions distributed largely around the hydrophilic 'kink' of the channel (*Figure 3—figure supplement 1*). The *Drosophila* $Q_9$ headgroup is bound 11 and 14 Å away, respectively, from its proposed ligating partners NDUFS2-His97 and NDUFS2-Tyr146, between the '1F' site described in *Sus scrofa* complex I (*Gu et al., 2022*) and the '$Q_m$' site described in *Escherichia coli* (*Kravchuk et al., 2022*). The wide spectrum of headgroup positions identified in different complex I structures suggests that substrates may shuttle in a step-wise manner, occupying numerous sites of localised energy minima (*Chung et al., 2021*; *Hoias Teixeira and Menegon Arantes, 2019*; *Warnau et al., 2018*). Thus, although the site observed here is clearly separated from the reactive site, it is only broadly defined and not a highly specific site.

## Modified domain disposition between the *Drosophila* and mammalian active states

In the mammalian complex, two subunits, NDUFA5 on the hydrophilic domain and NDUFA10 on the membrane domain, meet in the corner of the L-shape forming an interface between the two domains. Upon deactivation of the mammalian enzyme, the altered disposition of the hydrophilic and membrane domains changes the NDUFA5/NDUFA10 interface and decreases their contact area (*Agip et al., 2018*; *Zhu et al., 2016*). The nature and extent of the interface thereby provide an easy way to evaluate the active/deactive status of mammalian structures. Both subunits are present in the *Drosophila* enzyme, and *Figure 4a* compares their relative positions in the *Dm*1 active state to their relative positions in the active and deactive mammalian states. With the structures aligned to subunit NDUFA10, subunit NDUFA5, which is dominated by a three-helix bundle, clearly lies in an intermediate position in the *Dm*1 state, it does not overlay its position in the mammalian active state. The N-terminus of hydrophilic-domain subunit NDUFS2 that runs along the top of the membrane domain is also in an intermediate position. However, the NDUFA5/NDUFA10 contact area still matches closely to that observed in the mammalian active state (388 Å$^2$ vs. 354 and 131 Å$^2$ in the bovine active and deactive states [*Chung et al., 2022b*], respectively), and the NDUFA5/NDUFA10 interface is clearly different as it incorporates the N-terminus of the NDUFS4 subunit (residues 33–49; *Figure 4b*), which is substantially extended relative to in mammalian species (*Figure 4c*). Contacts between the NDUFS4 N-terminal 'tether' and subunits NDUFA5 and NDUFA10 of 182 and 427 Å$^2$, respectively, further stabilise the interface and, by extension, the relative disposition of the hydrophilic and membrane domains in the *Drosophila Dm1* active state.

Further comparison of the mammalian and *Drosophila* active-state structures showed that subunits NDUFS2, NDUFS7, and ND1 that constitute the ubiquinone-binding site (*Figure 3*), overlay closely (RMSD 0.574 between *Dm*1 and the bovine active state [PDB ID: 7QSK]; *Chung et al., 2022b*) but that the structures then diverge along the membrane domain, shifting the position and orientation of subunit ND2 (*Figure 5a*). As NDUFA5 is bound to NDUFS2 and NDUFA10 to ND2, their relative positions thus also change. Within the connecting subdomain between ND1 and ND2 that contains subunits ND3, ND6, and ND4L, the arrangement of the TMHs in subunit ND6 (*Figure 5b*) clearly differs between the *Drosophila* and mammalian enzymes (despite them both containing a fully α-helical ND6-TMH3). In particular, ND6-TMH4 is markedly displaced, enhancing a cleft between ND6 and ND1 in which three phospholipid molecules are observed (*Figure 1—figure supplement 4*). The position of ND6-TMH4 is remarkably variable between different species and states of complex I, suggesting that it is not functionally important (*Figure 5—figure supplement 1*). Structures around ND6-TMH4 also vary (*Figure 5b and c*): (i) the ND6-TMH4–5 loop is restructured and its β-hairpin disrupted by neighbouring ND4L-TMH1, (ii) ND6-TMH1 is displaced away from TMH4 to avoid steric clashes, (iii) the ND6-TMH3–4 loop is restructured to accommodate the movement of TMH4, and (iv) the C-terminal loop of NDUFA9, located just above the reordered ND6-TMH3–4 loop, is retracted. The ND3-TMH1–2 loop, which is also adjacent to the restructured region, remains ordered with ND3-Cys41 occluded (*Figure 5c*). Strikingly, the residues of the central axis that link the terminus of the E-channel to the start of the first antiporter-like subunit (ND2) are not affected by the altered connecting subdomain structure, which thus adjusts the relative disposition of subunits in the hydrophilic and membrane domains in the *Drosophila Dm*1 state *without* affecting the catalytic machinery of the active state structure.

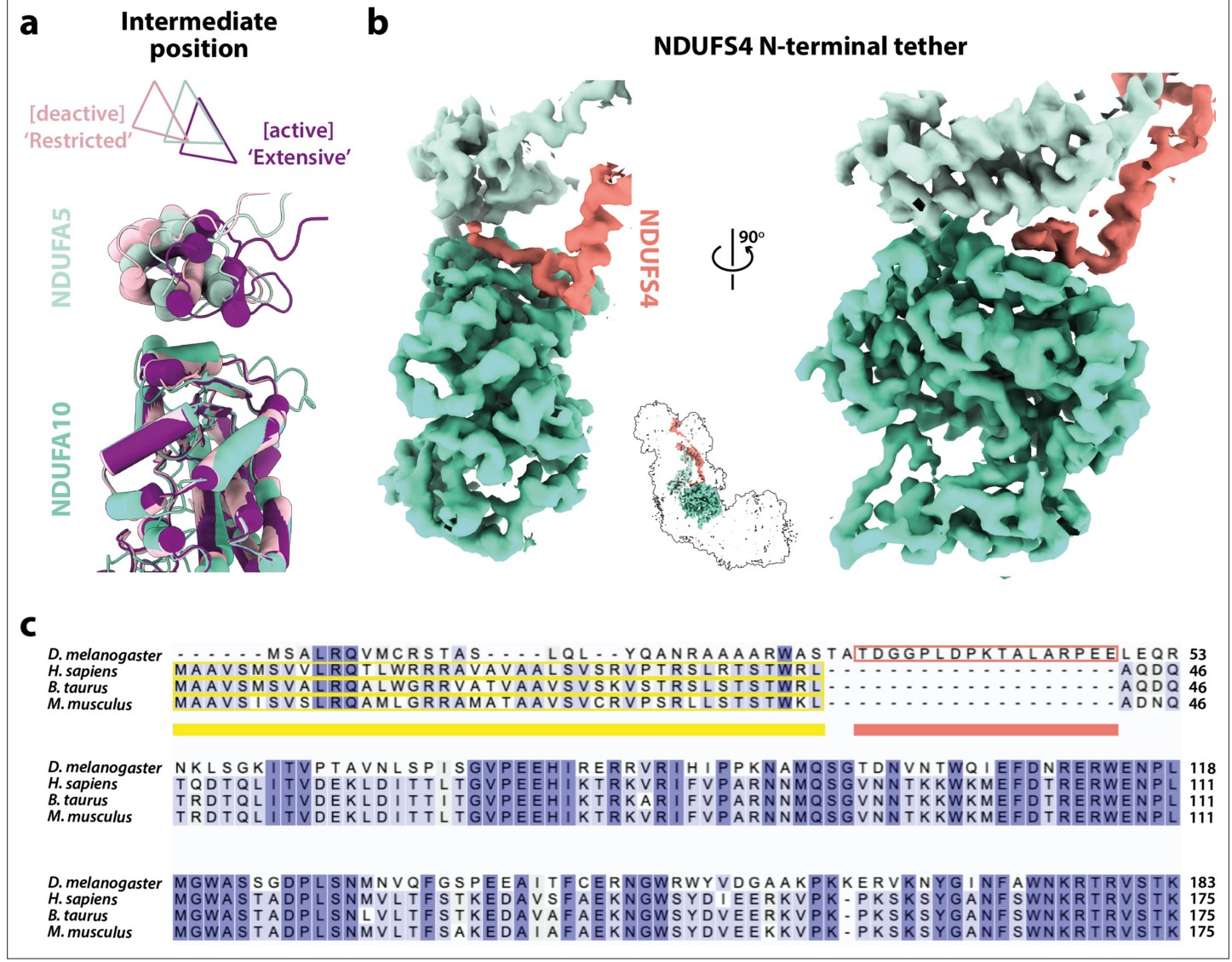

**Figure 4.** The domain interface between subunits NDUFA5, NDUFA10, and NDUFS4 in *Drosophila* complex I. (**a**) The interface between subunits NDUFA5 (mint) and NDUFA10 (turquoise) of *Drosophila* complex I is compared against the active (purple; PDB ID: 7QSK) and deactive (light pink; PDB ID: 7QSM) states of bovine complex I (*Chung et al., 2022b*), displaying an 'intermediate' conformation. Triangles indicate positions of the three-helix bundles in NDUFA5. The structures are aligned to subunit NDUFA10. (**b**) The extended N-terminal loop of NDUFS4 (salmon) specific to *Drosophila* complex I is tethered between NDUFA5 and NDUFA10, locking them in place. Inset shows the positions of the three subunits in complex I. The active-state (*Dm*1) *Drosophila* complex I map is shown at a threshold of 0.013 in *UCSF ChimeraX* (*Pettersen et al., 2021*). (**c**) Sequence alignment of NDUFS4 across a selection of NDUFA5/NDUFA10-containing organisms. Residues are coloured by similarity. Known mitochondrial targeting sequences are highlighted in yellow, and the modelled N-terminal extension of NDUFS4 in the *Dm*1 active-state structure is highlighted in salmon. UniProt IDs used for the alignment in *Clustal Omega 1.2.4* (*Sievers et al., 2011*): *Drosophila melanogaster*, Q9VWI0, *Homo sapiens*, O43181, *Bos taurus*, Q02375, *Mus musculus*, Q9CXZ1.

## Minor states with restricted deactive characteristics

Inspection of the set of local features in the core subunits (*Agip et al., 2018*; *Blaza et al., 2018*; *Chung et al., 2022a*; *Chung et al., 2022b*; *Letts et al., 2019*; *Zhu et al., 2016*) that differentiate the mammalian active and deactive states in the two minor states revealed that both the *Dm*2 and *Dm*3 states also most closely correspond to the mammalian active state. *Figure 6a* shows that, in the *Dm*2 state, two features of the mammalian deactive state are present: a π-bulge has formed in ND6-TMH3 and ND1-TMH4-Tyr149 has 'flipped' its conformation (*Chung et al., 2022b*; *Grba and Hirst, 2020*; *Kampjut and Sazanov, 2020*). However, all the other key elements remain in their active

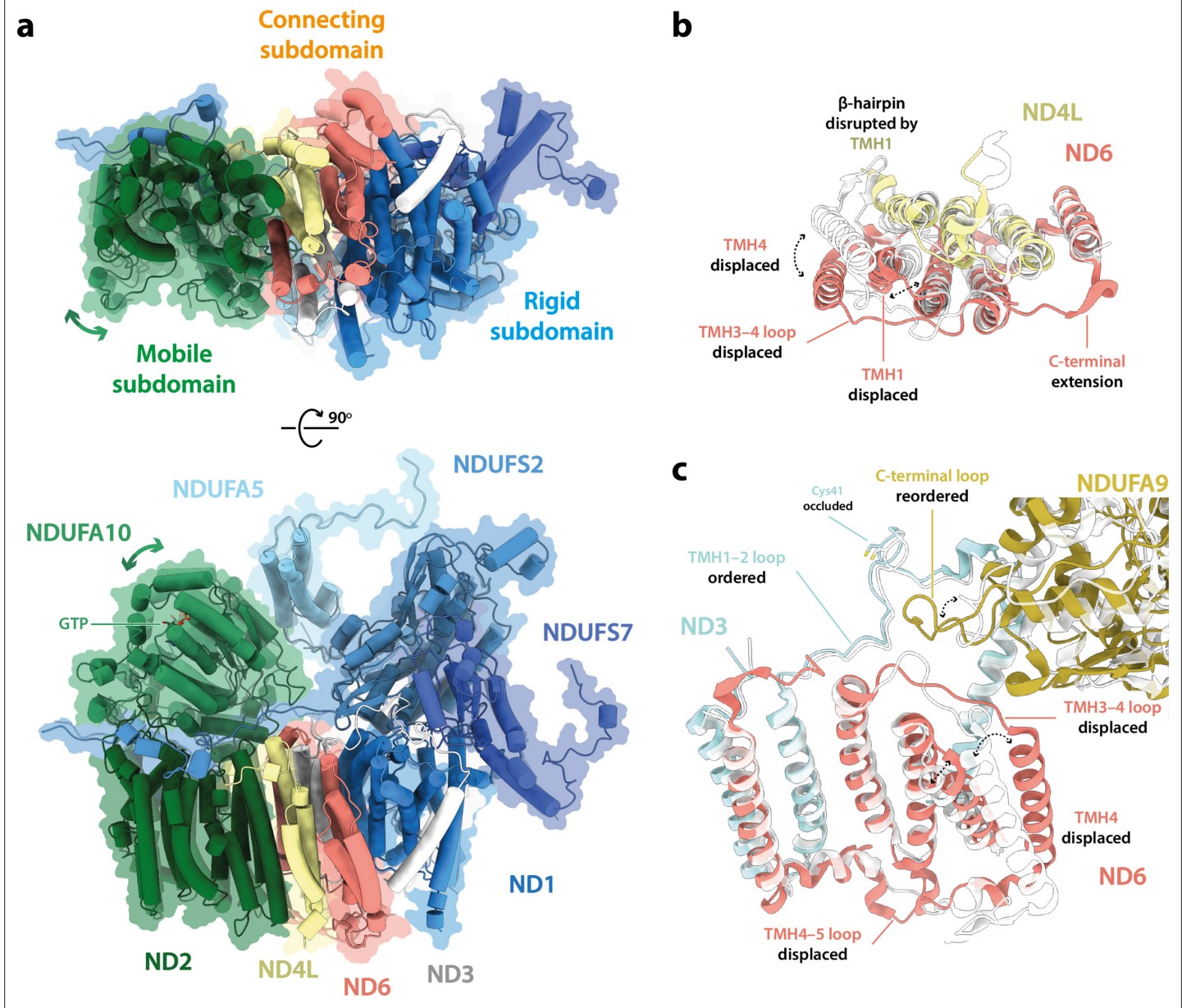

**Figure 5.** The structure of subunit ND6 and the connecting subdomain between subunits ND1 and ND2 alters the relative domain dispositions in the *Drosophila* active state relative to the mammalian active state. (**a**) The structures of subunits NDUFS2, NDUFS7, NDUFA5, and ND1 are tightly conserved between the *Drosophila* (*Dm*1, solid cartoon) and mammalian (PDB ID: 7QSK [*Chung et al., 2022b*], transparent cartoon) active states, forming a rigid subdomain. Subunits ND3, ND6, and ND4L form a connecting subdomain that differs, shifting the position and orientation of the mobile subdomain containing subunits ND2 and NDUFA10. The altered connecting domain changes the domain interface between NDUFA5 and NDUFA10. The models for the *Drosophila* and mammalian complexes are aligned on subunit NDUFS2 and shown alongside a flat surface representation of the *Drosophila* model. (**b, c**) Changes to the structure of the ND6 subunit, plus structural changes in adjacent subunits. The models for the *Drosophila* (*Dm*1, coloured) and mammalian (PDB ID: 7QSK [*Chung et al., 2022b*], white) active states are overlaid on subunit ND6.

The online version of this article includes the following figure supplement(s) for figure 5:

**Figure supplement 1.** Comparison of the position of ND6-TMH4 in the *Dm*1 state with the positions of ND6-TMH4 in other structures.

states, including ND1-TMH4, which remains in its bent conformation, and as a result the ubiquinone-binding site remains enclosed and sealed from the matrix (*Figure 6b and c*). The same is true for the *Dm*3 state (*Figure 6—figure supplement 1*). Notably, the trigonal junction between ND3-Cys41, NDUFS2-His93, and ND1-Tyr134 (*Grba and Hirst, 2020*) is preserved in all three states, occluding the Cys from the matrix and explaining why a mammalian-type deactive state of *Drosophila* complex

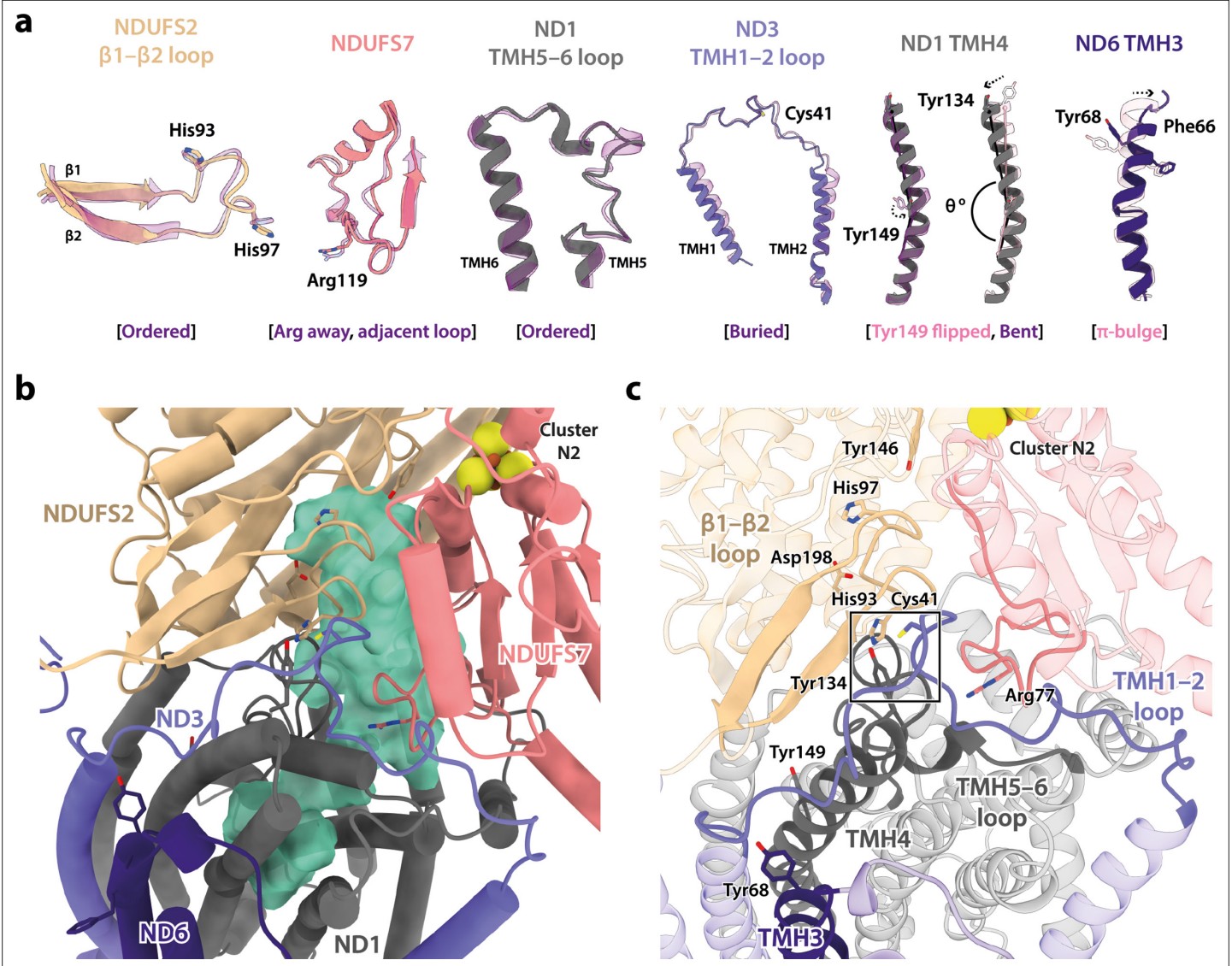

**Figure 6.** Local structural elements show that the *Dm2* state of *Drosophila* complex I most closely resembles the mammalian active state, with only two deactive-like features in the membrane domain. (**a**) The local elements in the core subunits, individually compared against an active-state bovine structure (transparent purple; PDB ID: 7QSK) (***Chung et al., 2022b***), show that all ubiquinone-binding site features of the *Dm2* complex are in the active state, whereas in the membrane domain ND1-Tyr149 and ND6-TMH3 match the bovine deactive state (transparent pink; PDB ID: 7QSM) (***Chung et al., 2022b***). Active state-specific key local features are indicated in square brackets in purple and deactive state-specific features in pink. The same features are highlighted and labelled in (**b**) and (**c**), respectively, showing that subunits NDUFS2, NDUFS7, ND1, ND3, and ND6 encapsulate a fully structured and sealed Q-binding cavity (aquamarine surface; detected by *CASTp; Tian et al., 2018*). The box in (**c**) denotes the trigonal junction (ND3-Cys41, NDUFS2-His93, and ND1-Tyr134).

The online version of this article includes the following figure supplement(s) for figure 6:

**Figure supplement 1.** Local structural elements in the *Dm2* state are conserved in *Dm3*.

I could not be trapped in biochemical assays (***Figure 2—figure supplement 1***). We conclude that, in contrast to the mammalian enzyme, *Drosophila* complex I does not form an 'open' resting state, it rests only in 'closed' conformations with the ubiquinone-binding site enclosed and sealed from the matrix.

Further comparison of the *Dm*1 and *Dm*2 states (***Figure 7a and b***) shows that the 'twisting' motion that relates their global conformations originates in changes in the 'connecting subdomain' between the rigid and mobile subdomains described in ***Figure 5a***, where the deactive-like elements of *Dm*2 are located (the π-bulge in ND6-TMH3 and ND1-TMH4-Tyr149). The twisting motion displaces the

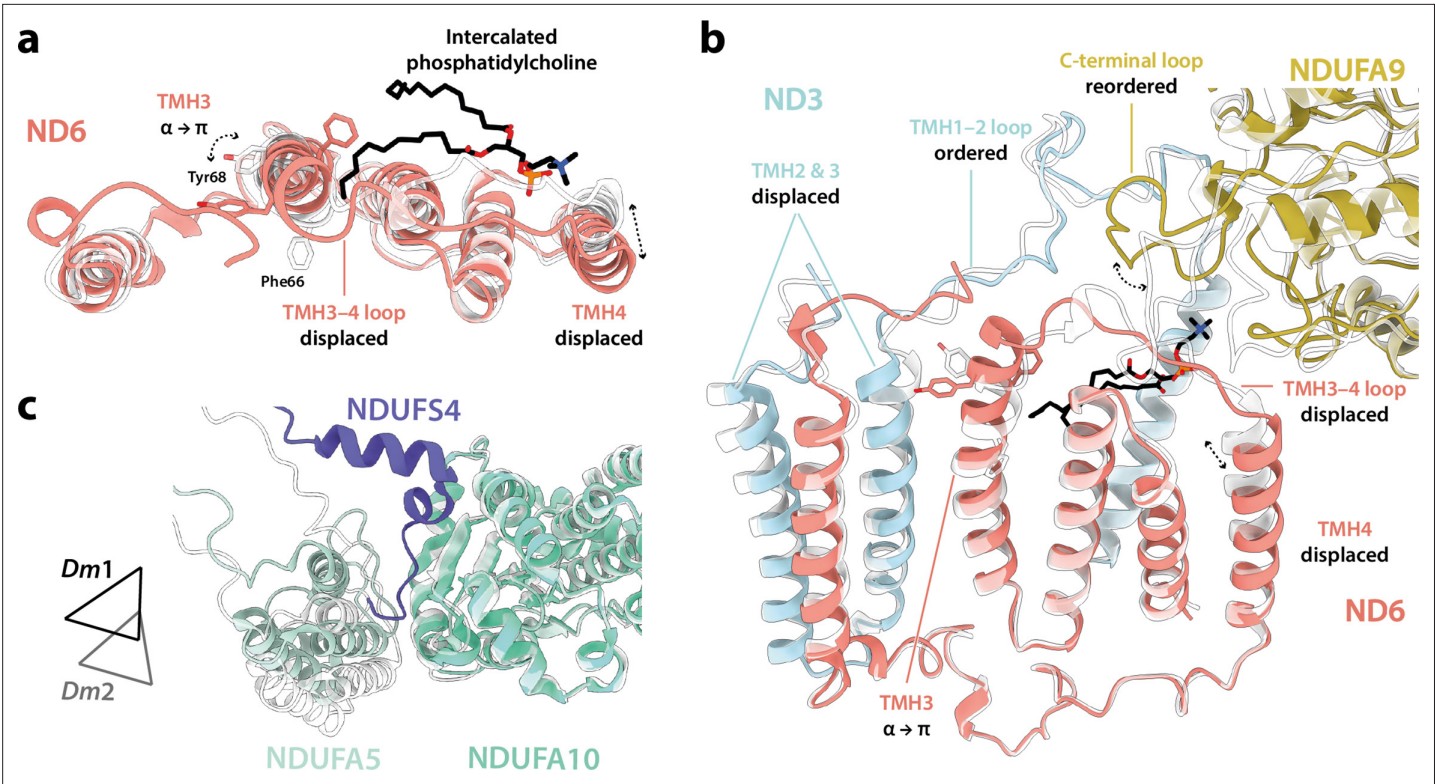

**Figure 7.** Differences between the *Dm*1 and *Dm*2 states at the connecting subdomain and at the domain interface between subunits NDUFA5, NDUFA10, and NDUFS4. (**a, b**) Changes to the structure of the ND6 subunit, plus structural changes in adjacent subunits. The intercalated phosphatidylcholine molecule is present in the *Dm*1 state only. (**c**) The interface between subunits NDUFA5 (mint), NDUFA10 (turquoise), and NDUFS4 (slate) of the *Dm*1 state is compared against *Dm*2 (white). Triangles indicate positions of the three-helix bundles in NDUFA5. The N-terminal NDUFS4 tether occupies the interface in *Dm*1 only, and the NDUFA5/NDUFA10 interface area is decreased in *Dm*2. The models for the *Dm*1 (coloured) and *Dm*2 (white) states are overlaid on subunit ND6 in (**a**) and (**b**), and on subunit NDUFA10 in (**c**).

N-terminus of subunit NDUFS2 (adjacent to ND6-TMH5) and changes the NDUFA5/NDUFA10 interface, displacing and disordering the N-terminus of subunit NDUFS4 (*Figure 7c*) and causing a small decrease in the interface area (from 388 Å² in *Dm*1 to 333 Å² in *Dm*2, relative to 354 and 131 Å² in the active and deactive states of bovine complex I; *Chung et al., 2022b*). Inspection of the region around the π-bulge in ND6-TMH3 revealed a further striking change between the *Dm*1 and *Dm*2 states. In *Dm*1, the tail of a phosphatidylcholine molecule is intercalated into the structure, sterically obstructing the rotation of bulky residues on ND6-TMH3 around the helical axis to form the π-bulge (*Figure 7a and b*). It is absent from the *Dm*2 state (and also from *Dm*3, where the local protein conformation matches *Dm*2; *Figure 6—figure supplement 1*). In the *Dm*1 state, the phosphatidylcholine headgroup stabilises the ND6-TMH3–4 loop at the top of ND6-TMH4, whereas its absence in *Dm*2 allows the TMH3–4 loop and TMH4 to move, along with a further adjustment to the adjacent C-terminal loop of NDUFA9 and displacement of ND3-TMH2–3. The lipid may either have been ejected during relaxation of ND6-TMH3 into a π-bulge structure, or removed during detergent extraction, promoting π-bulge formation.

## Discussion

The structures determined here for *Drosophila* complex I confirm its close relationships with mammalian complex I and thus its potential as a powerful genetically tractable model system for studying mammalian-specific aspects of complex I biology. The close-to-identical subunit compositions and structures of the mammalian and *Drosophila* enzymes now enable genetic approaches to be applied to elucidate, for example, the roles of the supernumerary subunits, the assembly pathway, and the detrimental effects of clinically identified pathological point mutations. Importantly, these aspects

can be studied in physiologically relevant in vivo environments and in specific cell types and tissues, extending the scope of earlier studies in cultured mammalian cells (*Guerrero-Castillo et al., 2017*; *Stroud et al., 2016*). However, our structures also reveal limitations in *Drosophila* as a model organism for complex I, as the *Drosophila* enzyme, despite its remarkable similarity to the mammalian enzyme, does not undergo the full mammalian-type active/deactive transition. Our cryo-EM analyses revealed the major class of enzyme particle (*Dm*1) in the active resting state, with all the characteristics of the mammalian active state enzyme. Two minor states (*Dm*2 and *Dm*3) also more closely resemble the active state, and we were unable to either detect a mammalian-type deactive resting state in biochemical assays, or to generate one by incubation of the enzyme at 37°C (the method used to deactivate mammalian complex I). However, we note that our biochemical assay relies on the availability of ND3-Cys41, just one characteristic that distinguishes the mammalian active and deactive states; the functional consequences of conversion to the *Dm*2 state are currently unknown, most notably whether it is (like the mammalian active state) able to catalyse RET, or (like the mammalian deactive state) unable to do so.

Comparison of the *Dm*1 (active) and *Dm*2 (twisted) structures of *Drosophila* complex I determined here suggests that the *Dm*2 state is a relaxed state, which may be considered a structurally curtailed form of the full mammalian-type deactive transition (*Agip et al., 2018*; *Blaza et al., 2018*; *Chung et al., 2022a*; *Chung et al., 2022b*; *Zhu et al., 2016*). In changes that also occur in the mammalian transition, a π-bulge forms in ND6-TMH3, and the nearby sidechain of ND1-TMH4-Tyr149 flips in conformation. Although water molecules cannot be resolved in our structures, these changes are expected to alter the connectivity between the E-channel and the central axis of charged residues along the membrane domain as reported previously for mammalian, yeast, and bacterial species (*Agip et al., 2018*; *Blaza et al., 2018*; *Chung et al., 2022b*; *Grba and Hirst, 2020*; *Kampjut and Sazanov, 2020*; *Kravchuk et al., 2022*; *Parey et al., 2021*). Furthermore, the limited and local changes we observe in the ND6 region result in a limited twisting of the global conformation in the *Dm*2 state, a motion that qualitatively resembles (but to a much lesser extent) the twisting of the deactive enzyme.

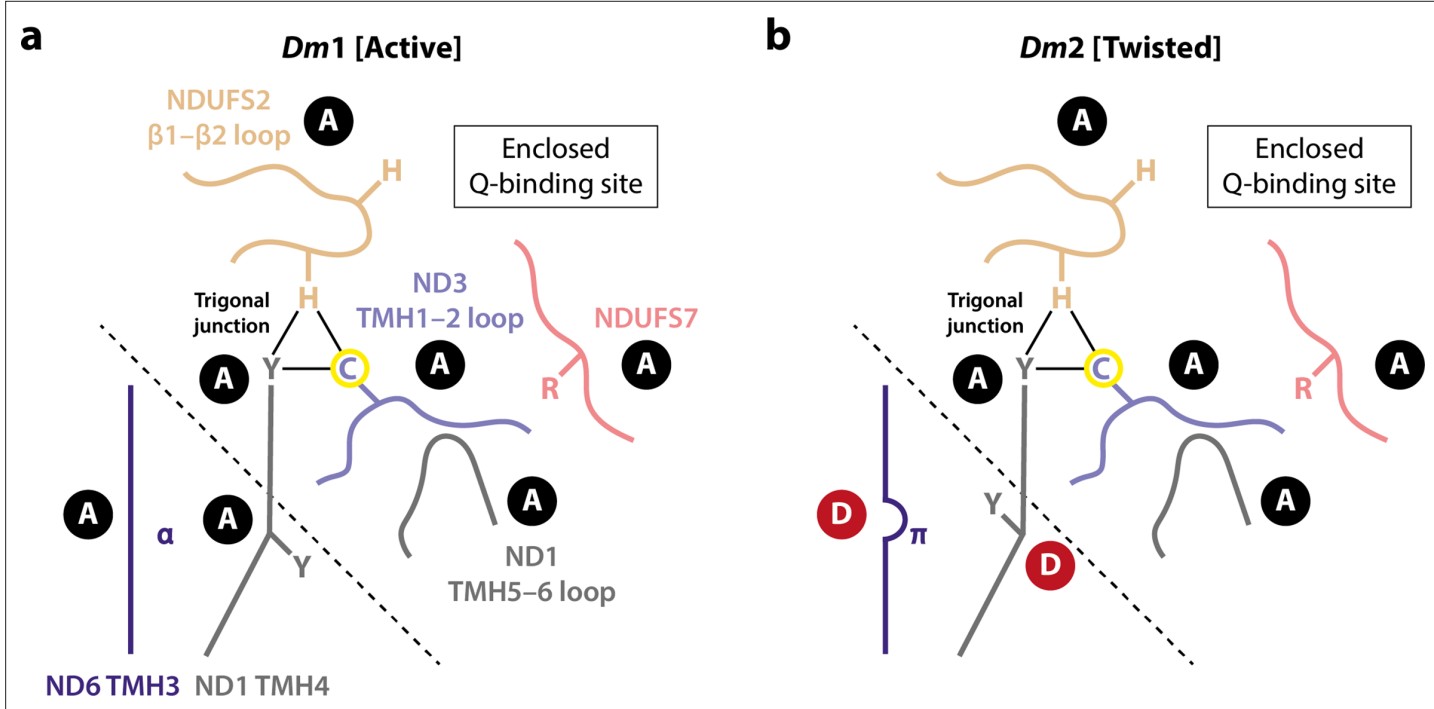

**Figure 8.** Schematic representation of the status of local active/deactive elements in the (**a**) *Dm*1 and (**b**) *Dm*2 states of *Drosophila* complex I. Local elements that change conformation in the mammalian active/deactive transition are shown and labelled as A for active and D for deactive, respectively. In the *Dm*2 state, ND6-TMH3 and ND1-TMH4-Tyr149 are in the D conformation. The boundary of the A and D regions is marked with a dashed line. In the mammalian deactive state, the top section of ND1-TMH4 moves, straightening the helix and resulting in loss of the trigonal junction (ND3-Cys41, NDUFS2-His93, and ND1-Tyr134), destructuring of the NDUFS2, ND3, and ND1 loops and restructuring of the NDUFS7 loop and NDUFS7-Arg119. ND3-Cys41, the derivatisable marker of the deactive state in mammalian/eukaryotic complex I (*Galkin et al., 2008*), is indicated with a yellow circle.

However, the cascade of changes that also occurs in the mammalian-type deactive transition does not follow: ND1-TMH4 does not straighten its conformation, the trigonal junction between ND3-Cys41, NDUFS2-His93, and ND1-Tyr134 is preserved, and so the conformational change from ND6-TMH3 does not propagate to the ubiquinone-binding site, which remains fully ordered, sealed from the matrix, and in its active state (*Figure 6*). This lack of direct correlation between the status of the α-helix/π-bulge and the structuring of the ubiquinone-binding site (*Figure 8*) argues against a concerted and structurally enforced connection between them being crucial for catalysis (*Kampjut and Sazanov, 2022*; *Kampjut and Sazanov, 2020*; *Kravchuk et al., 2022*). Although structures of complex I from other (non-mammalian) species have also been reported with a π-bulge in ND6-TMH3 but without the 'opening' of the ubiquinone-binding site observed in the mammalian deactive state (*Chung et al., 2022a*), our *Drosophila Dm*2 structure is the first example in which the π-bulge and a fully ordered, active ubiquinone-binding site have been observed together. Our structures are consistent with the elements that change during the mammalian deactive transition (such as the π-bulge) being mobile during catalysis, but do not suggest that they move in a coherent and coordinated transition, with the ubiquinone-binding site open to the matrix, during catalysis.

The observation, together, of the active *Dm*1 state and the 'curtailed-deactive' *Dm*2 state raises two questions: what causes the π-bulge to form in *Dm*2, and why does the conformational cascade to the mammalian-type deactive state not occur in the *Drosophila* enzyme (*Figure 8*)? First, it is possible that delipidation of the complex during detergent extraction removes the intercalated phospholipid that obstructs π-bulge formation in the *Dm*1 state, allowing conversion to *Dm*2. However, a similar intercalated phospholipid has not been observed in any mammalian active-state structure, so it may only bind when catalysis stops, or be an artefact of enzyme purification. Indeed, if ND6-TMH3 converts between its π-bulge and α-helical structures during catalysis (*Agip et al., 2018*; *Kampjut and Sazanov, 2020*; *Kravchuk et al., 2022*; *Parey et al., 2021*; *Röpke et al., 2021*), then the intercalating phospholipid is very unlikely to be present in the α-helical state, moving repeatedly in and out. Alternatively, it is possible that enzyme twisting, induced by loss of the NDUFS4 tether from the NDUFA5/NDUFA10 interface during purification, causes the π-bulge to form: this possibility may be addressed in future by genetic truncation of the NDUFS4 tether from the N-terminus of the mature subunit. Second, if formation of the π-bulge in *Drosophila* represents a curtailed-deactive transition, then conversion to a full mammalian-type deactive state would be accompanied by further twisting, disruption of the NDUFA5/NDUFA10 interface, and destructuring of the ubiquinone-binding site. That these changes are not observed in *Drosophila* complex I is likely due to the modified domain disposition in the *Dm*1 state that is stabilised by the structure of the connecting subdomain and accommodating changes in linked structures such as the NDUFA5/NDUFA10 interface. We propose that the stable domain disposition is resistant to further twisting and so resists the local changes that accompany it in the mammalian deactive transition. Computational simulations of the *Dm*1 structure may help further elucidate the answer to this question in future. Notably, our proposal implies high activation energy barriers for the 'opening' of the ubiquinone-binding site to the matrix in the *Drosophila* enzyme, arguing against opening and closing of the site during catalysis (*Kampjut and Sazanov, 2020*; *Kravchuk et al., 2022*).

The *Dm*3 'cracked' state is not discussed in detail here as we suspect it is an artefact resulting from detergent-induced loss of stability in the distal membrane domain of the *Dm*2 state. Similar opening and relaxation of the ND2–ND4 interface has also been observed in the 'slack' state of bovine complex I (*Chung et al., 2022b*; *Zhu et al., 2016*), as well as in a catalytically inactive state of complex I from rhesus macaque (*Agip et al., 2019*), and in pronounced open states of the ovine complex (*Kampjut and Sazanov, 2020*). In all cases, opening of the ND2–ND4 interface is linked to loss of density for nearby subunit NDUFA11, and to changes in the C-terminal section of the ND5 transverse helix and anchor helix (*Figure 2—figure supplement 2*). It may result from delipidation during enzyme purification, most likely removal of phospholipids from the interface on both sides of the complex, including 'behind' the transverse helix (*Figure 1—figure supplement 4b*). Consistent with this picture, treatment of the mammalian enzyme with zwitterionic detergents or prolonged incubation in detergent solution leads to fractionation at this interface (*Hirst et al., 2003*; *Zhu et al., 2015*).

The deactive transition and RET are linked in mammalian complex I biology, as deactivation protects against the burst of ROS production that occurs upon reperfusion by RET (RET-ROS), driven by oxidation of the reduced succinate pool that accumulates during ischaemia, leading to IR injury

(*Dröse et al., 2016*; *Galkin and Moncada, 2017*; *Wright et al., 2022*; *Yin et al., 2021*). The deactivation of complex I minimises the RET-ROS burst and tissue damage upon reperfusion because the deactive state of mammalian complex I is unable to catalyse RET (*Kotlyar and Vinogradov, 1990*; *Wright et al., 2022*; *Yin et al., 2021*). An elegant demonstration is provided by the ND6-P25L variant of mouse complex I, which deactivates much more rapidly than the wild-type enzyme, preventing RET-ROS catalysis and thereby protecting against IR injury (*Yin et al., 2021*). Strikingly, while *Drosophila* do not appear to adopt a mammalian-type deactive state, they are able to survive long periods of hypoxia followed by reoxygenation (*Haddad, 2006*; *Zhou and Haddad, 2013*), raising the question of whether they are protected by a corresponding mechanism. ROS production by RET has been described in studies of *Drosophila* mitochondria (although not demonstrated directly in the isolated enzyme) (*Scialò et al., 2016*), and the ability of *Drosophila* complex I to catalyse RET is consistent with it persisting in the active state (*Dm*1) when catalysis stops, rather than deactivating. Alternative mechanisms are therefore required to explain the resistance of *Drosophila* to hypoxia−reoxygenation challenges, such as greater robustness to oxidative stress from a RET-ROS-induced stress-responsive transcriptional programme (*Scialò et al., 2020*) and/or metabolic adaptations (*Perkins et al., 2012*). Future genetic studies that exploit structural insights will illuminate these mechanisms and provide new perspectives on the mechanisms of mammalian complex I.

## Materials and methods

**Key resources table**

| Reagent type (species) or resource | Designation | Source or reference | Identifiers | Additional information |
|---|---|---|---|---|
| Gene (*Drosophila melanogaster*) | NDUFA3, Dmel gene *CG9034* | FlyBase | UniProt ID: Q9W380 FlyBase ID: FBgn0040931 | |
| Biological sample (*D. melanogaster*) | isogenic *w*[1118] | Bloomington Drosophila Stock Center | RRID:BDSC_6326 | |
| Chemical compound, drug | Fatty acid-free bovine serum albumin | Merck Millipore | CAS number: 9048-46-8 | |
| Chemical compound, drug | EDTA-free cOmplete protease inhibitor cocktail | Roche | COEDTAF-RO | |
| Chemical compound, drug | Dodecyl-β-D-maltoside (DDM) | Merck Milipore | CAS number: 69227-93-6 | |
| Chemical compound, drug | Asolectin from soy bean | Avanti | CAS number: 8030-76-0 | |
| Chemical compound, drug | CHAPS (3-((3-cholamidopropyl) dimethylammonio)−1-propanesulfonate) | Calbiochem | CAS number: 75621-03-3 | |
| Chemical compound, drug | NADH | Merck Millipore | CAS number: 606-68-8 | |
| Chemical compound, drug | Decylubiquinone (dQ) | Merck Millipore | CAS number: 55486-00-5 | |
| Chemical compound, drug | *N*-ethylmaleimide (NEM) | Merck Millipore | CAS number: 128-53-0 | |
| Chemical compound, drug | 11-Mercaptoundecyl hexaethylene glycol | SensoPath Technologies | SPT-0011P6 | |
| Software, algorithm | EPU | Thermo Fisher Scientific | | |
| Software, algorithm | RELION-3.0 | *Zivanov et al., 2018* | RRID:SCR_016274 | |
| Software, algorithm | RELION-3.1 | *Zivanov et al., 2020* | RRID:SCR_016274 | |
| Software, algorithm | MotionCor2 | *Zheng et al., 2017* | RRID:SCR_016499 | |
| Software, algorithm | CTFFIND-4.1 | *Rohou and Grigorieff, 2015* | RRID:SCR_016732 | |

*Continued on next page*

*Continued*

| Reagent type (species) or resource | Designation | Source or reference | Identifiers | Additional information |
|---|---|---|---|---|
| Software, algorithm | crYOLO 1.5.3 | *Wagner et al., 2019* | https://cryolo.readthedocs.io/en/stable/ | |
| Software, algorithm | UCSF ChimeraX | *Pettersen et al., 2021* | RRID:SCR_015872 | |
| Software, algorithm | 3DFSC | *Tan et al., 2017* | https://github.com/LyumkisLab/3DFSC; *LyumkisLab, 2019* version 3.0 | |
| Software, algorithm | SWISS-MODEL | *Waterhouse et al., 2018* | RRID:SCR_018123 | |
| Software, algorithm | UCSF Chimera | *Pettersen et al., 2004* | RRID:SCR_004097 | |
| Software, algorithm | MODELLER | *Webb and Sali, 2016* | RRID:SCR_008395 | |
| Software, algorithm | Coot | *Casañal et al., 2020* | RRID:SCR_014222 | |
| Software, algorithm | ISOLDE | *Croll, 2018* | https://isolde.cimr.cam.ac.uk/ | |
| Software, algorithm | PyMOL 2.5.2 | *Schrodinger, 2022* | RRID:SCR_000305 | |
| Software, algorithm | Phenix 1.18.2–3874 | *Liebschner et al., 2019* | RRID:SCR_014224 | |
| Software, algorithm | MapQ | *Pintilie et al., 2020* | | |
| Software, algorithm | MolProbity | *Chen et al., 2010* | RRID:SCR_014226 | |
| Software, algorithm | EMRinger | *Barad et al., 2015* | | |
| Software, algorithm | CASTp | *Tian et al., 2018* | http://sts.bioe.uic.edu/castp/ | |
| Other | UltrAuFoil gold grids | Quantifoil; *Russo and Passmore, 2014* | | R 0.6/1 Gold foil on Gold 300 mesh grid |

## *Drosophila* stocks and husbandry

Flies of a common wild type-equivalent genotype, isogenic $w^{1118}$ (RRID:BDSC_6326), were obtained from Bloomington Drosophila Stock Center (RRID:SCR_006457), raised and kept under standard conditions in a temperature-controlled incubator with a 12 hr:12 hr light:dark cycle at 25°C and 65% relative humidity, on food consisting of agar, cornmeal, molasses, propionic acid, and yeast. Approximately 5500 mixed adults collected 5 days after eclosion were used for the preparation of the cryo-EM sample.

## Preparation of *Drosophila* complex I

All experimental procedures were carried at 4°C unless otherwise stated. One volume of $w^{1118}$ flies (1 mL is equivalent to ~100 flies) was mixed with five volumes of homogenisation buffer containing 20 mM Tris-HCl pH 7.8, 250 mM sucrose, 2 mM EDTA, 2 mM EGTA, 1% (w/v) fatty acid-free bovine serum albumin (BSA, Merck) and 1× EDTA-free cOmplete protease inhibitor cocktail (Roche) (one tablet per 50 mL of buffer). Mitochondria were prepared using a differential centrifugation method. Briefly, 20 mL of fly suspension were homogenised by 10 strokes with a motor-driven Teflon pestle at 1300 rpm in a 30 mL Wheaton glass homogeniser. The homogenate was then centrifuged at 1000 × *g* for 5 min, and the supernatant filtered through a muslin cloth to remove cuticles. The same process was repeated a second time. Then, mitochondria were pelleted at 3000 × *g* for 10 min and washed with 5 mL of homogenisation buffer but without BSA. Finally, mitochondria were collected by centrifugation at 7000 × *g* for 10 min and resuspended in 5.8 mL (56 mg of protein) of resuspension buffer containing 20 mM Tris-HCl (pH 7.8), 20% glycerol (v/v), 2 mM EDTA, 2 mM EGTA, 1% and 1× EDTA-free cOmplete protease inhibitor cocktail (one tablet per 50 mL of buffer). Isolated mitochondria were stored at –80°C until further use. Mitochondrial membranes were prepared as described previously for mouse samples (*Agip et al., 2018*). Defrosted mitochondria were diluted to 5 mg mL$^{-1}$ in resuspension buffer then ruptured on ice with a Q700 Sonicator (Qsonica) in three intervals (5 s bursts each followed by a 30 s pause) at an amplitude setting of 65%. Membranes (38 mg of protein)

were collected by centrifugation at 75,000 × *g* for 1 hr, then resuspended to 4.9 mg mL$^{-1}$ in the same buffer and stored at –80°C.

Purification of *Drosophila melanogaster* complex I followed the same procedure as previously described for mouse complex I, with minor adjustments (*Agip et al., 2018*). While being continuously stirred on ice, mitochondrial membranes (7.6 mL at 4.9 mg mL$^{-1}$) were solubilised for 30 min by the drop-wise addition of dodecyl-β-D-maltoside (DDM) to a final concentration of 0.75% from a 10% stock solution. The solubilised membranes were then centrifuged at 48,000 × *g* for 30 min and the clarified supernatant loaded on to a Hi-Trap Q HP anion exchange column (1 mL; Cytiva) pre-equilibrated with elution buffer A (20 mM Tris-HCl pH 7.8°C, 2 mM EDTA, 2 mM EGTA, 0.1% DDM, 10% ethylene glycol [v/v, VWR], 0.005% asolectin [Avanti], and 0.005% CHAPS [Calbiochem]) and operated at a flow rate of 0.3 mL min$^{-1}$. The column was washed with several column volumes of buffer A until the 280 nm absorbance reached the baseline. Unwanted proteins were eluted with seven column volumes of 20% buffer B (buffer A + 1 M NaCl), then complex I was eluted with an additional seven column volumes of 35% buffer B. Fractions containing complex I (ca. 3 mL) were pooled and concentrated to ca. 100 μL using an Amicon-Ultra filter device (100 kDa molecular weight cut-off; Amicon, Millipore). The concentrated sample was then injected onto a Superose 6 Increase size exclusion column (150 × 5 mm; Cytiva) pre-equilibrated in buffer C (20 mM Tris-HCl pH 7.8°C, 150 mM NaCl, and 0.05% DDM) operated at a flow rate of 0.03 mL min$^{-1}$. All chromatographic procedures described were carried out using an ÄKTA micro FPLC system (Cytiva) with elution monitored at 280 and 420 nm. Complex I concentrations were estimated at 280 nm ($\varepsilon$ = 0.2 mg mL$^{-1}$ mm$^{-1}$). The total collected protein was estimated at 0.6 mg, and the peak concentration was 3.4 mg mL$^{-1}$.

## Kinetic activity measurements

All activity measurements were measured on a 96-well Spectramax 384 plate reader at 32°C. For NADH:decylubiquinone (dQ) oxidoreductase activities, NADH (200 μM final concentration) was used to initiate catalysis by complex I (0.2 μg mL$^{-1}$) with 200 μM dQ, 0.15% (w/v) asolectin, and 0.15% (w/v) CHAPS in 20 mM Tris-HCl (pH 7.55). NADH oxidation was monitored at 340–380 nm ($\varepsilon$ = 4.81 mM$^{-1}$ cm$^{-1}$), and was confirmed to be sensitive to rotenone and piericidin A. The cryo-EM sample had an activity of 7.3 ± 0.3 μmol min$^{-1}$ mg$^{-1}$ (mean ± SD; n = 4).

For evaluation of the active/deactive state ratio of *Drosophila* complex I using the *N*-ethylmaleimide (NEM) assay (*Galkin et al., 2008*; *Yin et al., 2021*), 4 mg mL$^{-1}$ mitochondria were incubated with 2 mM NEM or the equivalent volume of DMSO on ice for 20 min., before determining the NADH:O$_2$ oxidoreductase activity. The mitochondria had been frozen for storage before measurement. To attempt to deactivate the complex, the mitochondria were incubated at 37°C for 30 min (equivalent to, or longer than, the treatments used to deactivate complex I in mammalian mitochondrial membranes [*Agip et al., 2018*; *Blaza et al., 2018*]). NADH:O$_2$ oxidoreductase activities were measured in 20 mM Tris-HCl (pH 7.55) using 10 μg mL$^{-1}$ mitochondria and 10 μg mL$^{-1}$ alamethicin, and initiated using 200 μM NADH. NADH oxidation was monitored as described above.

## Cryo-EM grid preparation and image acquisition

UltrAuFoil gold grids (0.6/1, Quantifoil) (*Russo and Passmore, 2014*) were prepared for *Drosophila* complex I as described previously (*Blaza et al., 2018*). First, the grids were glow discharged with plasma under vacuum for 90 s at 20 mA then incubated for 7 days under anaerobic and room temperature conditions in a solution of 5 mM 11-mercaptoundecyl hexaethylene glycol (SPT-0011P6, SensoPath Technologies) (*Meyerson et al., 2015*). Grids were then washed several times in ethanol and left to dry. Complex I (3.4 mg mL$^{-1}$) was then applied (3 μL per grid) to the treated grids in a Vitrobot Mark IV (Thermo Fisher Scientific) set to 4°C and 100% relative humidity. Grids were blotted for 10 s with a force setting of –10, before being plunged into liquid ethane. Frozen grids were then stored in liquid nitrogen before screening and data collection.

Both cryo-EM screening and high-resolution image collection were carried out on a Titan Krios (Thermo Fisher Scientific) at University of Cambridge cryo-EM facility. The Titan Krios microscope for data collection was operating at an accelerated voltage of 300 kV and equipped with a Gatan K2 detector utilising a GIF quantum energy filter with a slit width of 20 eV. The microscope was operated in electron counting mode with a nominal sampling rate of 1.07 Å pix$^{-1}$ (nominal magnification of 130,000) and a dose rate of ca. 4.18 electrons Å$^{-2}$ s$^{-1}$. The specimen was radiated for 10 s over 40

frames with a total exposure amounting to ca. 42 electrons Å⁻². A 100 µm and 50 µm objective and C2 aperture, respectively, were inserted during high-resolution imaging. The microscope was operated with EPU software and the defocus range was set to –1.0 to –2.0 µm, with an autofocus routine run every 5 µm.

## Cryo-EM data processing

All 3082 collected movies were subjected to processing by *RELION-3.0* and *3.1* (*Zivanov et al., 2020*; *Zivanov et al., 2018*) except where stated otherwise. Micrographs were motion-corrected using *MotionCor2* (*Zheng et al., 2017*) with 5 × 5 patches, and contrast transfer function (CTF) parameters estimated using *CTFFIND-4.1* (*Rohou and Grigorieff, 2015*) in *RELION-3.0*. In parallel, the motion-corrected micrographs were exported and subjected to particle autopicking using a general model without training in *crYOLO 1.5.3* (*Wagner et al., 2019*), resulting in 194,538 picked particles. Ice-contaminated micrographs were removed to give 180,342 particles from 2852 micrographs. Particles were extracted with an initial downscaling to 6.0 Å pixel⁻¹ (box size of 80) and subjected to initial 2D and 3D classification steps to remove junk particles, yielding a total of 93,332 particles. These particles were re-extracted at the nominal pixel size of 1.07 Å pixel⁻¹ (box size of 450) and used to reconstruct a 3.74 Å resolution map using the 3D autorefinement procedure in *RELION*, at the calibrated pixel size of 1.048 Å pixel⁻¹ (*Spikes et al., 2020*). The active state map of mouse complex I (EMD-4345) (*Agip et al., 2018*) was used as a reference map for the 3D reconstruction. Bayesian polishing (*Zivanov et al., 2019*) was then applied and CTF parameters, including astigmatism, defocus, and beam tilt, estimated using the CTF refinement procedure in *RELION-3.0*. Particles were subjected to additional rounds of classifications, to further remove junk and bad complex I particles. From hereon, all data processing was performed in *RELION-3.1*, at the nominal pixel size of 1.07 Å pixel⁻¹, then corrected to the calibrated pixel size of 1.048 Å pixel⁻¹ at the postprocessing or local resolution stages. The particles were subject to iterative rounds of CTF refinement (*Zivanov et al., 2020*), to estimate aniso-tropic magnification, beam tilt, trefoil, fourth-order aberration, and per-particle defocus, astigmatism and *B*-factor parameters. Particles with an *rlnNrOfSignificantSamples* value greater than 3000 were removed to give 65,864 particles. Using a complex I mask (generated from a working model using *RELION MaskCreate*) and with solvent flattening, the global resolution of the 3D refined map was 3.23 Å (according to a gold-standard Fourier shell correlation [FSC] of 0.143; *Rosenthal and Henderson, 2003*). 3D classification (number of classes, $K = 5$, local angular search to 0.2° sampling) was then performed, and three complex I classes, *Dm*1, *Dm*2, and *Dm*3, were identified and retained, containing 37,608, 12,343, and 13,520 particles, respectively, a ratio of roughly 3:1:1. Clear cryo-EM densities for *Dm*2-specific local features, including the 'flipped' ND1-TMH4-Tyr149 and the ND6-TMH3 π-bulge, revealed no evidence for *Dm*1 contamination in the *Dm*2 population. The 'Focus-Revert-Classify' classification strategy (*Letts et al., 2019*), applied using the regularisation parameter $t = 8$ and $K = 5$, yielded comparable population distributions (three complex I classes matching *Dm*1, *Dm*2, and *Dm*3, plus two junk classes) whilst 3D classification without alignment using $t = 20$ and $K ≤ 12$ yielded two <4 Å complex I classes, with the major class matching *Dm*1 and the minor class an apparent mixture of *Dm*2 and *Dm*3. The 3D classification approach with local angular sampling was therefore employed to give the final set of *Dm*1, *Dm*2, and *Dm*3 particles as described above. Using model-generated (*Dm*1 and *Dm*2) or map-generated (*Dm*3) masks and solvent-flattening, the three classes refined to 3.28, 3.68, and 3.96 Å resolution, respectively. Global resolutions were estimated from two independent half maps using a gold-standard FSC of 0.143 (*Rosenthal and Henderson, 2003*) in *RELION postprocess*. The final map was globally sharpened (or blurred) in *RELION postpro-cess* using user-provided *B*-factor values. The model-generated or map-generated mask used for 3D refinement procedures and resolution estimation was generated in *UCSF ChimeraX* (*Pettersen et al., 2021*) using the *molmap* (*Dm*1 and *Dm*2) or *vop threshold* (*Dm*3) functions, before being low-pass filtered to 15 Å and having a 6-pixel soft cosine edge added using *RELION MaskCreate*. Local reso-lution was estimated using *RELION LocRes*. Mollweide projections were plotted using *Python* and *Matplotlib*, and the degree of directional resolution anisotropy calculated using the *3DFSC* program suite (*Tan et al., 2017*).

## Model building, refinement, and validation

Model coordinates were built into the 3.3 Å resolution *Dm*1 *Drosophila* complex I map, with a published mouse complex I structure (PDB ID: 6G2J) (*Agip et al., 2018*) serving as a homology model.

*SWISS-MODEL (Waterhouse et al., 2018)* was used to generate an initial model for each subunit and the models rigid-body fitted into the map using *Chimera (Pettersen et al., 2004)*. *MODELLER (Webb and Sali, 2016)* was used to generate models for subunits NDUFB6, NDUFB9, and NDUFB8 as the homology models were unsatisfactory. The sequence for the *Drosophila* NDUFA3 subunit (UniProt ID: Q9W380; Dmel gene *CG9034*; FlyBase ID: FBgn0040931) was identified in routine peptide mass spectrometry analyses of the purified enzyme. The evidence for Q9W380 relies on a single peptide (LGYVVYR, 9.1% coverage) but the MASCOT score is 43, well above 30 (the 99% confidence limit), and inspection of the distribution of tryptic cleavage sites suggests no further peptides would be expected to be detected. Models for subunits NDUFA2 and NDUFC1 were deleted due to the lack of corresponding densities in the cryo-EM maps, and N- and C-terminal extensions were built where necessary. It was noted that densities for the N-termini of ND1 and ND5 were extended beyond the reviewed UniProt sequences (P18929 and P18932, respectively). Therefore, to incorporate the correct translation start site, UniProt IDs C7DZL9 and C7DZL4 (*Stewart and Beckenbach, 2009*) were used to build models for subunits ND1 and ND5, respectively. UniProt ID A0A024E3A5 was used for subunit NDUFA9 to include a L174F mutation supported by the *Drosophila* cryo-EM density features. Notably, the N-terminus of core subunit NDUFS7 is resolved for the first time in *Drosophila* complex I (*Figure 1—figure supplement 5*). Preliminary model building and real-space refinements were carried out in *Coot 0.9-pre (Casañal et al., 2020)*. Then GPU-powered *ISOLDE 1.0 (Croll, 2018)*, which implements a molecular dynamic approach to model refinement, was used to iterate through the model, improving the map-to-model fit, resolving clashes and maintaining good protein stereochemistry. Emerging modelling errors were monitored using a real-time validation functionality present in *ISOLDE* and corrected. Densities for existing and additional phospholipid molecules were identified with the *Unmodelled blobs* tool in *Coot 0.9.6.2 (Casañal et al., 2020)*. All non-cardiolipin phospholipids were modelled as phosphatidylethanolamines, the largest component of the phospholipid composition of the *Drosophila* mitochondrial membranes (*Jones et al., 1992*), unless density features indicated phosphatidylcholine to be more likely. Lipid tails were clipped where necessary using the *delete* tools in *Coot* and *PyMOL 2.5.2 (Schrodinger, 2022)*. dGTP was modelled in subunit NDUFA10 (*Molina-Granada et al., 2022*). The model was then *Curlew* all-atom-refined using *Coot* and real-space refined against the active-state map using *phenix.real_space_refine* in *Phenix 1.18.2–3874 (Liebschner et al., 2019)* with custom geometry restraints. Ligand restraints were generated using *Phenix eLBOW*. No secondary structure restraints were used during real-space refinement of the *Dm*1 model. The model was checked manually in *Coot*, new resolvable regions built, and rotameric and/or Ramachandran outliers corrected. Atom resolvabilities (*Q*-scores) were calculated using *MapQ* (*Pintilie et al., 2020*) and any persisting outliers identified and corrected. The model was then real-space refined in *Phenix* as described above to produce the final *Dm*1 model.

To build the *Dm*2 model, the *Dm*1 *Drosophila* model was rigid-body fitted into the *Dm*2 map using the *Fit in map* tool in *UCSF ChimeraX (Pettersen et al., 2021)* followed by rigid-body fitting by subunit in *Phenix 1.18.2–3874*, and *Curlew* all-atom-refined using *Coot 0.9.6.2*. The *Dm*2 model was manually inspected and new resolvable regions, less resolved regions, and/or conformationally different regions built or deleted manually in *Coot*, and locally refined in *ISOLDE 1.4 (Croll, 2018)*. *Q*-scores were calculated using *MapQ*, and any outliers identified and corrected. Existing lipids were checked against their densities and deleted where appropriate; lipid tails were similarly clipped where necessary as described above. The model was then real-space refined against the *Dm*2 map in *Phenix 1.18.2–3874* with custom geometry restraints and secondary structure restraints (identified by *ksdssp*). Iteratively, rotameric and Ramachandran outliers were corrected manually in *Coot* and real-space refined in *Phenix*. The final real-space refinement for the *Dm*2 model was performed without secondary structure restraints in *Phenix*. The model statistics for the *Dm*1 and *Dm*2 classes (*Table 1*) were produced by *Phenix*, *MolProbity (Chen et al., 2010)*, and *EMRinger (Barad et al., 2015)*. Model-to-map FSC curves were generated using *phenix.validation_cryoem* in *Phenix*.

Individual subunits from the *Dm*2 model were rigid-body fitted into the *Dm*3 map in *UCSF ChimeraX (Pettersen et al., 2021)* to generate a tentative model for the *Dm*3 state for visualisation.

## Cryo-EM model analyses

RMSD calculations between models were performed using the *Align* command in *PyMOL 2.5.2 (Schrodinger, 2022)*. Buried surface area between subunits were calculated using the *measure buriedarea*

command in *UCSF ChimeraX* (*Pettersen et al., 2021*). The interior surface of the ubiquinone-binding channel was predicted using *CASTp* (*Tian et al., 2018*), which computes a protein surface topology from a PDB model. The default 1.4 Å radius probe was used and the results were visualised in *PyMOL* using the *CASTpyMOL 3.1* plugin and by *UCSF ChimeraX*.

## Acknowledgements

We thank D Chirgadze (University of Cambridge Cryo-EM facility) for assistance with grid screening and cryo-EM data collection; T Croll (Cambridge Institute for Medical Research) for assistance with *ISOLDE* and I M Fearnley and S Ding (MRC MBU) for mass spectrometry analyses. This work was supported by the Medical Research Council (MC_UU_00015/6 and MC_UU_00028/6 to AJW and MC_UU_00015/2 and MC_UU_00028/1 to JH). *Drosophila* were obtained from the Bloomington Drosophila Stock Center, which is supported by grant NIH P40OD018537.

## Additional information

### Funding

| Funder | Grant reference number | Author |
|---|---|---|
| Medical Research Council | MC_UU_00015/6 | Alexander J Whitworth |
| Medical Research Council | MC_UU_00028/6 | Alexander J Whitworth |
| Medical Research Council | MC_UU_00015/2 | Judy Hirst |
| Medical Research Council | MC_UU_00028/1 | Judy Hirst |

The funders had no role in study design, data collection and interpretation, or the decision to submit the work for publication.

### Author contributions

Ahmed-Noor A Agip, Conceptualization, Data curation, Formal analysis, Investigation, Methodology, Writing - review and editing; Injae Chung, Data curation, Formal analysis, Validation, Investigation, Visualization, Writing - original draft, Writing - review and editing; Alvaro Sanchez-Martinez, Investigation, Methodology, Writing - review and editing; Alexander J Whitworth, Conceptualization, Supervision, Funding acquisition, Project administration, Writing - review and editing; Judy Hirst, Conceptualization, Formal analysis, Supervision, Funding acquisition, Investigation, Writing - original draft, Project administration, Writing - review and editing

### Author ORCIDs

Ahmed-Noor A Agip http://orcid.org/0000-0002-3020-8262
Injae Chung http://orcid.org/0000-0002-2902-4677
Alvaro Sanchez-Martinez http://orcid.org/0000-0002-2728-6251
Alexander J Whitworth http://orcid.org/0000-0002-1154-6629
Judy Hirst http://orcid.org/0000-0001-8667-6797

### Decision letter and Author response

Decision letter https://doi.org/10.7554/eLife.84424.sa1
Author response https://doi.org/10.7554/eLife.84424.sa2

## Additional files

### Supplementary files
• MDAR checklist

## Data availability

Structural data have been deposited in the EMDB and PDB databases under the following accession codes: EMD-15936 and 8B9Z (Dm1; active), EMD-15937 and 8BA0 (Dm2; twisted), and EMD-15938 (Dm3; cracked).

The following datasets were generated:

| Author(s) | Year | Dataset title | Dataset URL | Database and Identifier |
|---|---|---|---|---|
| Agip A-NA, Chung I, Sanchez-Martinez A, Whitworth AJ, Hirst J | 2022 | *Drosophila melanogaster* complex I in the Active state (*Dm*1) | https://www.rcsb.org/structure/8B9Z | RCSB Protein Data Bank, 8B9Z |
| Agip A-NA, Chung I, Sanchez-Martinez A, Whitworth AJ, Hirst J | 2022 | *Drosophila melanogaster* complex I in the Active state (*Dm*1) | https://www.ebi.ac.uk/emdb/EMD-15936 | EMDataResource, EMD-15936 |
| Agip A-NA, Chung I, Sanchez-Martinez A, Whitworth AJ, Hirst J | 2022 | *Drosophila melanogaster* complex I in the Twisted state (*Dm*2) | https://www.rcsb.org/structure/8BA0 | RCSB Protein Data Bank, 8BA0 |
| Agip A-NA, Chung I, Sanchez-Martinez A, Whitworth AJ, Hirst J | 2022 | *Drosophila melanogaster* complex I in the Twisted state (*Dm*2) | https://www.ebi.ac.uk/emdb/EMD-15937 | EMDataResource, EMD-15937 |
| Agip A-NA, Chung I, Sanchez-Martinez A, Whitworth AJ, Hirst J | 2022 | *Drosophila melanogaster* complex I in the Cracked state (*Dm*3) | https://www.ebi.ac.uk/emdb/EMD-15938 | EMDataResource, EMD-15938 |

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
