## [Editor Report]

This important article advances our understanding of respiratory complex I. The cryoEM data are convincing and the interpretation of different conformational states will stimulate discussions in the field. The work introduces *Drosophila melanogaster* as a model organism to study respiratory complex I and will be of interest to researchers studying respiratory enzymes, the evolution of respiration and mitochondrial diseases.

---

## [Decision Letter]

**Decision letter after peer review:**

Thank you for submitting your article "Cryo-EM structures of mitochondrial respiratory complex I from *Drosophila melanogaster*" for consideration by *eLife*. Your article has been reviewed by 3 peer reviewers, and the evaluation has been overseen by a Reviewing Editor and Volker Dötsch as the Senior Editor. The following individuals involved in review of your submission have agreed to reveal their identity: Yongchan Lee (Reviewer #2); Hans-Peter Braun (Reviewer #3).

Essential revisions:

The reviewers requested to address a few issues to improve the manuscript. I would like to highlight and add a few points here.

1. The activity of the Dm complex I preparation is somewhat hidden in the Materials and methods section. Please report the inhibitor sensitive activity in the text. To allow better comparison with other preparations please give the value in µmol min-1 mg-1 and electrons s^-1^.

2. The NEM assay alone might not be sufficient to exclude that the A/D transition is completely absent in Dm complex I, especially because Dm2 is considered to be a relaxed or deactive-like state. Please show the traces of activity measurements or report if a lag phase exists or not. The A/D transition should be also tested by monitoring the impact of divalent cations when added before or after start of the activity measurement.

3. Reviewer 1 pointed out that the assignments of Dm1 and Dm2 need more discussion. This is a critical point. Dm2 shows two features of complex I in the deactive state, but you exclude that Drosphila complex I undergoes an A/D transition. These two statements are difficult to reconcile. Isn´t that at least suggesting that the pi-bulge of TMH3 of ND6 is actually not a hallmark of the D state? Please discuss.

4. It is discussed that in Dm2, a lack of transmission from the pi-bulge to the Q binding site argues against a direct link between the two being crucial for catalysis. This appears not fully justified because very likely the water chain connecting Q tunnel and hydrophilic axis is interrupted. It is clear that at the given resolution this cannot be observed here, but maybe you want to add a sentence on this point.

*Reviewer #1 (Recommendations for the authors):*

– The authors describe Dm1 as the active resting state. What makes it resting? How would it differ from a 'turnover' state?

– Also Dm2 structurally resembles the active state of the mammalian Complex I, but with some characteristic of the deactive form, such as the pi-bulge but not the ND3 loop or NDUFS2. Why is Dm2 not the active resting state?

– I believe that general reader might find the discussion of active / deactive / deactive-type / opening / twisted confusing and would benefit from a clarification in the Discussion section.

– What interactions stabilize the quinone headgroup in the modelled position? Are homologous residues present in the other isoform (with different resolved quinone positions)?

– "there is no apparent opening or closing of the angle between the domains" – could the authors estimate the angles to show that they are the same?

– Purification: could DDM bind to the quinone cavity and lead to partial inhibition of the activity or block expected conformational changes? I believe that this has been seen in other Complex I structures. Did the authors characterize the activity in other detergents?

– Previous data indicate that cardiolipin binds to Complex I, but I could not see any modeled cardiolipin molecules. Could the authors please comment on this?

Related to this point, the method section say: "The methods say all non-cardiolipin molecules were modelled", but there is no discussion of the cardiolipin molecules.

– Motion around NDUFA5/10 – the authors discuss that these subunits have different conformation in the mammalian active and deactive states, but it is not evident to me why it is expected that these subunits would affect e.g. the twisting shown in Figure 2?

– Activity: for the general reader, please provide a comparison to activities for example for the mammalian enzyme in the active and deactive states. Is the measured activity sensitive to known Complex I inhibitors?

– I understand that no water molecules could be resolved at this resolution, but the authors could comment on whether they find evidence that the conformational changes in Dm1 and Dm2 could lead to hydration changes and in this way affect the proton translocation?

*Reviewer #2 (Recommendations for the authors):*

– Would sample pre-treatment, such as heat and the addition of NADH or ubiquinone, change the ratio of Dm1 and Dm2 states, which might inform about the nature of these conformations?

---

## [Author Response]

Essential revisions:The reviewers requested to address a few issues to improve the manuscript. I would like to highlight and add a few points here.1. The activity of the Dm complex I preparation is somewhat hidden in the Materials and methods section. Please report the inhibitor sensitive activity in the text. To allow better comparison with other preparations please give the value in µmol min-1 mg-1 and electrons s^-1^.

We have amended the *Results section* to include this information on page 5: “The highest concentration peak fraction (3.4 mg ml^-1^), which exhibited an NADH oxidoreductase activity comparable to mammalian complex I of 7.3 ± 0.3 µmol min^-1^ mg^-1^ (*ca*. 120 NADH s^-1^) was collected and frozen…” As stated in *Materials and methods* we confirmed that the activity is sensitive to both rotenone and piericidin.

2. The NEM assay alone might not be sufficient to exclude that the A/D transition is completely absent in Dm complex I, especially because Dm2 is considered to be a relaxed or deactive-like state. Please show the traces of activity measurements or report if a lag phase exists or not. The A/D transition should be also tested by monitoring the impact of divalent cations when added before or after start of the activity measurement.

The NEM assay is only sufficient to conclude that Cys41 on the ND3-TMH1-2 loop is not exposed in *Drosophila* complex I, and that it does not become exposed upon heat treatments that are sufficient to deactive the mammalian enzyme. The NEM assay does not probe any other elements of the mammalian deactive transition, such as the ND6-TMH3 π-bulge. The NEM assay is useful for investigating the active/deactive status of the mammalian enzyme because there is a single transition between the two states, in which all the elements convert together. We agree that, unfortunately, the NEM assay is therefore not suitable for probing the conversion between *Dm*1 and *Dm*2 and we have now checked our manuscript to ensure that we have been clear on this point (notably, on page 12). We are aware, of course, that two other biochemical characteristics are associated with the mammalian deactive state. First, we have now revisited our assay traces for *Drosophila* complex I in mitochondria, membranes and isolated in detergent, and we can discern no clear evidence of any catalytic lag phase. Although we regard this as consistent with the fully formed ubiquinone-binding sites in all the *Dm*1, *Dm*2 and *Dm*3 states (no ‘refolding’ lag phase is required) we have not conducted sufficiently focussed investigations to make a definitive statement. Second, we have not yet carried out investigations of the effects of divalent cations on catalysis by *Drosophila* complex I. We are aware that divalent cations have been used successfully with complex I from *Yarrowia lipolytica*, but in our experiments on the mammalian enzyme we have not found this characteristic easy to reproduce or to distinguish from inhibition satisfactorily, and our experiments on the *Drosophila* enzyme were severely limited by the amount of material that was reasonably available. Unfortuantely, we do not currently have the resources available to produce the quantitites of enzyme that would be required for this study.

3. Reviewer 1 pointed out that the assignments of Dm1 and Dm2 need more discussion. This is a critical point. Dm2 shows two features of complex I in the deactive state, but you exclude that Drosphila complex I undergoes an A/D transition. These two statements are difficult to reconcile. Isn´t that at least suggesting that the pi-bulge of TMH3 of ND6 is actually not a hallmark of the D state? Please discuss.

As we have noted in our answer to reviewer 1, *Dm*1 contains all the features of the mammalian active resting state (Agip, 2018; Blaza et al., 2018; Chung et al., 2022b, 2022a; Zhu et al., 2016) and therefore we are confident in this assignment. *Dm*2 differs from *Dm*1 by the presence of a π-bulge in ND6-TMH3 and a flipped Tyr149 in ND1-TMH4, which are two features of the mammalian deactive resting state. However, the mammalian deactive resting state also exhibits a disordered ND3-TMH1-2 loop (i.e. exposed ND3-Cys39), disordered NDUFS2 β1-β2 loop, disordered ND1-TMH5-6 loop, bent ND1-TMH4 as well as a decreased interface area between NDUFA5 and 10 – none of these features are present in *Dm*2. *Dm*2 can therefore be considered intermediate between the mammalian active and deactive states, and we therefore chose to give it a different name (‘Twisted’). We described the Twisted state as a ‘restricted’ or ‘curtailed’ deactive state to attempt to capture the suggestion that it has partially converted to a mammalian-type deactive state. Whether *Dm*2 should be referred to as the ‘*Drosophila*-deactive state’ and whether we say *Dm*2 is formed in a ‘*Drosophila*-A/D transition’ is a matter of semantics. Furthermore, structural elements that move in the mammalian A/D transition, including the π-bulge, are also likely mobile during catalysis and so (although it is not our preferred explanation) we cannot discount that *Dm*2 is the result of pausing catalysis at a different point on the catalytic cycle than *Dm*1.

4. It is discussed that in Dm2, a lack of transmission from the pi-bulge to the Q binding site argues against a direct link between the two being crucial for catalysis. This appears not fully justified because very likely the water chain connecting Q tunnel and hydrophilic axis is interrupted. It is clear that at the given resolution this cannot be observed here, but maybe you want to add a sentence on this point.

We apologise that our text was not clear on this point. We intended to observe that the π-bulge can form without changing the structure of the ubiquinone-binding site, as demonstrated by comparison of our *Dm*1 and *Dm*2 structures. This is contrary to, for example, the mechanism proposed by Kravchuk et al. that requires a clean switch between two states (destructured Q-site elements plus π-bulge vs. structured and closed Q-site plus α-helix) in order to avoid the formation of the destructured:α-helix combination, which it is proposed would incur proton leak. Although we have not observed this particular state, our *Dm*2 state demonstrates that a clean switch between the two states described above is not structurally enforced. We have now amended our text on page 13 to be more clear on this point.

Reviewer #1 (Recommendations for the authors):– The authors describe Dm1 as the active resting state. What makes it resting? How would it differ from a 'turnover' state?

No substrates (NADH or quinone) were added to our cryo-EM sample so that the complex I is not undergoing catalysis and therefore is ‘at rest’. We use the term resting state to clearly differentiate the complex I structures in our preparation from the catalytic intermediates that have been described in the literature to be present in ‘turnover’ samples in the presence of substrates. As we describe, the active resting state refers to the ‘ready-to-go’ resting state; this is the terminology that is used throughout the complex I field.

– Also Dm2 structurally resembles the active state of the mammalian Complex I, but with some characteristic of the deactive form, such as the pi-bulge but not the ND3 loop or NDUFS2. Why is Dm2 not the active resting state?

Whereas *Dm*1 contains all the features of the mammalian active resting state (Agip, 2018; Blaza et al., 2018; Chung et al., 2022b, 2022a; Zhu et al., 2016), *Dm*2 deviates from it by the presence of a π-bulge in ND6-TMH3 and a flipped Tyr149 in ND1-TMH4. *Dm*2 is therefore not equivalent to the active resting state (or to the deactive resting state) and we therefore chose to give it a different name (‘Twisted’).

– I believe that general reader might find the discussion of active / deactive / deactive-type / opening / twisted confusing and would benefit from a clarification in the Discussion section.

It is unfortunate that nomenclature in the complex I field has recently become more confusing because different groups have opted to use different naming systems and have also made different assignments to biochemically characterised states. First, we have clearly described the active/deactive and closed/open states, and the relationships between them, in our introduction (paragraph 3). As noted above, our twisted state is intermediate between the active and deactive resting states. We realise that our use of ‘deactive-type state’ in the *Discussion* was confusing, and no longer use this term.

– What interactions stabilize the quinone headgroup in the modelled position? Are homologous residues present in the other isoform (with different resolved quinone positions)?

The quinone observed in the *Dm*1 structure, including its headgroup, is largely stabilised by hydrophobic interactions as indicated in Figure 3 —figure supplement 1. There are no other isoforms present in our structures. We note that no quinones were resolved in the *Dm*2 or *Dm*3 states, and that the region of the ubiquinone-binding site in question is predominantly populated by hydrophobic residues in all complex I structures.

– "there is no apparent opening or closing of the angle between the domains" – could the authors estimate the angles to show that they are the same?

‘Opening’ and ‘Closing’ are poorly defined descriptions of complex I conformational transitions that are widely used in the complex I field. The conformational transitions are in fact not represented by a single two dimensional angle but are the result of complicated three dimensional motion that, nevertheless, looks from the side like an opening and closing. We therefore opted for the term ‘apparent opening and closing’. To clarify this further we now no longer specifically refer to the angle between the domains.

– Purification: could DDM bind to the quinone cavity and lead to partial inhibition of the activity or block expected conformational changes? I believe that this has been seen in other Complex I structures. Did the authors characterize the activity in other detergents?

So far, DDM binding has only been observed in the ubiquinone-binding site of the deactive state in bovine (Chung et al., 2022b) and yeast (Grba and Hirst, 2020) complex I. Based on the cryo-EM densities observed in the ubiquinone-binding sites here, there is no clear evidence of DDM binding to any of the three *Drosophila* complex I structures. In addition, all three states contain a closed ubiquinone-binding site, and no DDM has been observed bound to a fully closed state. The question of whether DDM may bind to inhibit catalysis in other states (that are not present in our structural analysis) is hard to answer but the activity of our complex I purified in DDM was substantial (7.3 µmol min^-1^ mg^-1^), offering no support for it being inhibited. Please also see the comment below that specifically addresses the activity value.

– Previous data indicate that cardiolipin binds to Complex I, but I could not see any modeled cardiolipin molecules. Could the authors please comment on this?Related to this point, the method section say: "The methods say all non-cardiolipin molecules were modelled", but there is no discussion of the cardiolipin molecules.

Cardiolipin molecules have been modelled in the structure(s), but were not referred to in any particular detail because they were found in regions already observed in other complex I structures and not of obvious functional or catalytic significance. We have now indicated the modelled cardiolipins in Figure 1 —figure supplement 4.

– Motion around NDUFA5/10 – the authors discuss that these subunits have different conformation in the mammalian active and deactive states, but it is not evident to me why it is expected that these subunits would affect e.g. the twisting shown in Figure 2?

We assume that by ‘excepted’, the reviewer means ‘expected’. NDUFA5 and 10 form one of the key interfaces between the hydrophilic and membrane domains. NDUFA5 is part of the hydrophilic domain whereas NDUFA10 is part of the membrane domain. Due to the apparent closing/opening of the two domains, where they move relative to each other, in the mammalian active/deactive transition, the NDUFA5/10 interface and contact surface areas change. In *Drosophila*, between the active resting state and the twisted state, the hydrophilic and membrane domains also move relative to each other, and therefore the NDUFA5/10 interface and contact surface areas change. Interestingly, the structurally ordered N-terminus of NDUFS4 specific to *Drosophila* complex I stabilises the NDUFA5/10 interface in *Dm*1 as shown in Figure 4. Disordering of this N-terminal tether in *Dm*2 therefore disrupts this stabilised interface, assisting the hydrophilic and membrane domains twisting against each other as shown in Figure 2.

– Activity: for the general reader, please provide a comparison to activities for example for the mammalian enzyme in the active and deactive states. Is the measured activity sensitive to known Complex I inhibitors?

The activity of complex I from *Drosophila* (7.3 µmol min^-1^ mg^-1^) is comparable that of ovine complex I (5-6 µmol min^-1^ mg^-1^) (Kampjut and Sazanov, 2020; Letts et al., 2016), but lower than murine (10-12 µmol min^-1^ mg^-1^) (Agip et al., 2018) or bovine (20-24 µmol min^-1^ mg^-1^) (Chung et al., 2022b; Wright et al., 2022) complex I. We have amended the *Results section* to include this information on page 5: “The highest concentration peak fraction (3.4 mg ml^-1^), which exhibited an NADH oxidoreductase activity comparable to mammalian complex I of 7.3 ± 0.3 µmol min^-1^ mg^-1^ (*ca*. 120 NADH s^-1^), was collected and frozen…”. Due to very limited sample availability from fly preparations we were unable to conduct extensive inhibitor investigations, however, we did confirm that catalysis is sensitive to addition of rotenone and piericidin (now noted on page 18).

– I understand that no water molecules could be resolved at this resolution, but the authors could comment on whether they find evidence that the conformational changes in Dm1 and Dm2 could lead to hydration changes and in this way affect the proton translocation?

The formation of a π-bulge in ND6-TMH3 in the deactive state is known to block the connection between the E-channel and the central hydrophilic axis of charged residues, as reported previously by many structures of complex I, including mammalian, yeast, and bacterial species (Agip et al., 2018; Blaza et al., 2018; Chung et al., 2022b; Grba and Hirst, 2020; Kampjut and Sazanov, 2020; Kravchuk et al., 2022; Parey et al., 2021). In addition, ND1-TMH4-Tyr149 points into (mammalian active state) or away from (mammalian deactive state) the E-channel, affecting the connectivity between the ND1 cavity (extending from the ubiquinone-binding site) and the neighbouring ND1-Glu150 and ND3-Asp68 (towards ND3) (Chung et al., 2022b). Both features are conserved here, between *Dm*1 and *Dm*2, and therefore the same changes in hydration and proton connectivity are expected. We have now added a sentence on page 12 to note this: “Although water molecules cannot be resolved in our structures, these changes are expected to alter the connectivity between the E-channel and the central axis of charged residues along the membrane domain as reported previously for mammalian, yeast, and bacterial species (Agip et al., 2018; Blaza et al., 2018; Chung et al., 2022b; Grba and Hirst, 2020; Kampjut and Sazanov, 2020; Kravchuk et al., 2022; Parey et al., 2021).”.

Reviewer #2 (Recommendations for the authors):- Would sample pre-treatment, such as heat and the addition of NADH or ubiquinone, change the ratio of Dm1 and Dm2 states, which might inform about the nature of these conformations?

Indeed, it is possible that the ratio of *Dm*1 and *Dm*2 may change upon heating or addition of substrates, which may inform us of the biochemical relevance of these states. However, this is not within the scope of our study, which focuses only on the resting state enzymes. We agree that it will be interesting to pre-treat *Drosophila* complex I to change its biochemical status before performing single particle cryo-EM in our future studies.